# For self-supervised learning, Rationality implies generalization, provably

**Yamini Bansal**[*]
Harvard University

**Gal Kaplun**[*]
Harvard University

**Boaz Barak**[†]
Harvard University

## Abstract

We prove a new upper bound on the generalization gap of classifiers that are obtained by first using self-supervision to learn a representation $r$ of the training data, and then fitting a simple (e.g., linear) classifier $g$ to the labels. Specifically, we show that (under the assumptions described below) the generalization gap of such classifiers tends to zero if $\mathsf{C}(g) \ll n$, where $\mathsf{C}(g)$ is an appropriately-defined measure of the simple classifier $g$'s complexity, and $n$ is the number of training samples. We stress that our bound is *independent* of the complexity of the representation $r$.

We do not make any structural or conditional-independence assumptions on the representation-learning task, which can use *the same training dataset* that is later used for classification. Rather, we assume that the training procedure satisfies certain natural *noise-robustness* (adding small amount of label noise causes small degradation in performance) and *rationality* (getting the wrong label is not better than getting no label at all) conditions that widely hold across many standard architectures.

We also conduct an extensive empirical study of the generalization gap and the quantities used in our assumptions for a variety of self-supervision based algorithms, including SimCLR, AMDIM and BigBiGAN, on the CIFAR-10 and ImageNet datasets. We show that, unlike standard supervised classifiers, these algorithms display small generalization gap, and the bounds we prove on this gap are often non vacuous.

## 1 Introduction

The current standard approach for classification is "end-to-end supervised learning" where one fits a complex (e.g., a deep neural network) classifier to the given training set (Tan & Le, 2019; He et al., 2016). However, modern classifiers are heavily *over parameterized*, and as demonstrated by Zhang et al. (2017), can fit 100% of their training set even when given random labels as inputs (in which case test performance is no better than chance). Hence, the training performance of such methods is by itself no indication of their performance on new unseen test points.

In this work, we study a different class of supervised learning procedures that have recently attracted significant interest. These classifiers are obtained by: **(i)** performing pre-training with a self-supervised task (i.e., without labels) to obtain a complex representation of the data points, and then **(ii)** fitting a simple (e.g., linear) classifier on the representation and the labels. Such *"Self-Supervised + Simple"* (SSS for short) algorithms are commonly used in natural language processing tasks (Devlin et al., 2018; Brown et al., 2020), and have recently found uses in other domains as well (Ravanelli et al., 2020; Liu et al., 2019). Compared to standard "end-to-end supervised learning", SSS algorithms have several practical advantages. In particular, SSS algorithms can incorporate additional unlabeled data, the representation obtained can be useful for multiple downstream tasks, and they can have improved out-of-distribution performance (Hendrycks et al., 2019). Moreover, recent works show that even without additional unlabeled data, SSS algorithms can get close to state-of-art accuracy in several classification tasks (Chen et al., 2020b; He et al., 2020; Misra & Maaten, 2020;

---

[*]Equal contribution. Email: {ybansal, galkaplun}@g.harvard.edu
[†]Email: b@boazbarak.org

Tian et al., 2019). For instance, SimCLRv2 (Chen et al., 2020b) achieves $79.8\%$ top-1 performance on ImageNet with a variant of ResNet-152, on par with the end-to-end supervised accuracy of this architecture at $80.5\%$.

We show that SSS algorithms have another advantage over standard supervised learning—they often have a small *generalization gap* between their train and test accuracy, and we *prove non-vacuous bounds* on this gap. We stress that SSS algorithms use *over-parameterized* models to extract the representation, and reuse the *same training data* to learn a simple classifier on this representation. Thus, the final classifier they produce has high complexity by most standard measures, and it is by no means apriori evident that their generalization gap will be small.

Our bound is obtained by first noting that the generalization gap of *every* training algorithm is bounded by the sum of three quantities, which we name the **Robustness gap**, **Rationality gap**, and **Memorization gap** (we call this the **RRM bound**, see Fact I). We now describe these gaps at a high level, deferring the formal definitions to Section 2. All three gaps involve comparison with a setting where we inject *label noise* by replacing a small fraction $\eta$ of the labels with random values.

The *robustness gap* corresponds to the amount by which training performance degrades by noise injection. That is, it equals the difference between the standard expected training accuracy (with no label noise) and the expected training accuracy in the noisy setting; in both cases, we measure accuracy with respect to the original (uncorrupted) labels. The robustness gap is nearly always small, and sometimes provably so (see Section 3).

The *rationality gap* corresponds to the difference between performance on the noisy training samples (on which the training algorithm gets the wrong label) and test samples (on which it doesn't get any label at all), again with respect to uncorrupted labels. An optimal Bayesian procedure would have zero rationality gap, and indeed this gap is typically zero or small in practice. Since it is a non-standard quantity, We discuss the rationality gap in Section 3.1, and explain assuming it is small is both well-founded and does not trivialize the question of generalization.

The *memorization gap*, which often accounts for the lion's share of the generalization gap, corresponds to the difference in the noisy experiment between the training accuracy on the entire train set and the training accuracy on the samples that received the wrong label (both measured with respect to uncorrupted labels). The memorization gap can be thought of as quantifying the extent to which the classifier can "memorize" noisy labels, or act differently on the noisy points compared to the overall train set. The memorization gap is large in standard "end-to-end supervised training". In contrast, our main theoretical result is that for SSS algorithms, the memorization gap is small if the simple classifier has small complexity, *independently* of the complexity of the representation. As long as the simple classifier is under-parameterized (i.e., its complexity is asymptotically smaller than the sample size), our bound on the memorization gap tends to zero. When combined with small rationality and robustness, we get concrete non-vacuous generalization bounds for various SSS algorithms on the CIFAR-10 and ImageNet datasets (see Figures 1 and 4).

In a nutshell, our results are the following:

**Theoretical contributions.**

1. Our main theoretical result (Theorem II) is that the *memorization gap* of an SSS algorithm is bounded by $O(\sqrt{C/n})$ where $C$ is the complexity of the simple classifier produced in the "simple fit" stage. This bound is oblivious to the complexity of the representation produced in the pre-training and does not make any assumptions on the relationship between the representation learning method and the supervised learning task.

   One way to interpret this result is that we give a rigorous bound on the generalization gap of SSS algorithms, under the assumptions that the robustness and rationality gaps are bounded by some small constant (e.g., $5\%$). As mentioned below, these assumptions hold widely in practice across many different classifiers. Moreover, these assumptions are non-trivial and do not "assume away the difficulty". Indeed, there are many natural examples of training algorithms for which these assumptions hold but the generalization gap is large. Last, making some assumptions is *necessary* for a generalization bound to hold for SSS algorithms; see Remark 3.1 and Appendix E.

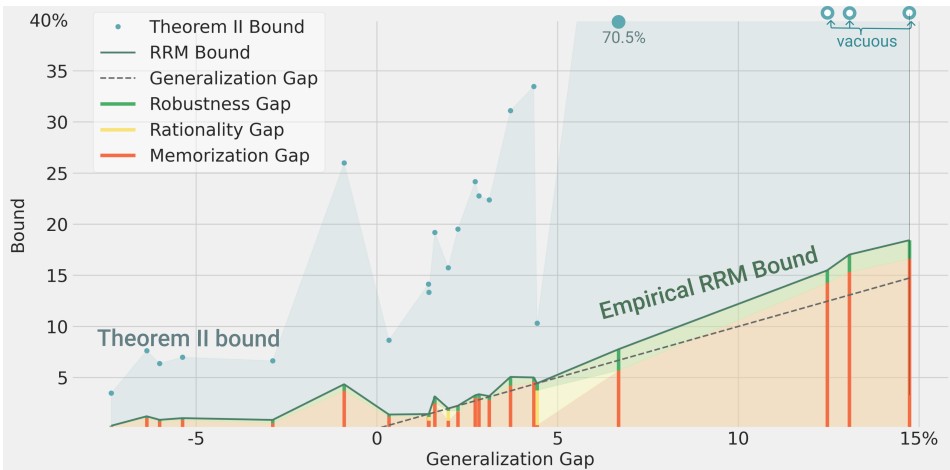

**Figure 1 – Empirical RRM bound.** The components of the RRM bound, as well as the upper bound of Theorem II for a variety of models on the CIFAR-10 dataset with noise $\eta = 0.05$.
Each vertical line corresponds to a single model (architecture + self-supervised task + fitting algorithm) and plots the RRM bound for this model. The green component corresponds to robustness, yellow to rationality, and red to memorization. The $x$ axis is the generalization gap, and so the RRM bound is always above the dashed $x = y$ line. A negative generalization gap can occur in algorithms that use augmentation. The blue dots correspond to the bound on the generalization gap obtained by replacing the memorization gap with the bound of Theorem II. See Sections 4 and D.3 for more information.

2. We also give a theoretical justification for the assumption of a small rationality gap, by proving that a positive rationality gap corresponds to "leaving performance on the table", in the sense that we can transform a learning procedure with a large rationality gap into a procedure with better test performance (Theorem 3.2).

**Empirical contributions.** We complement the theoretical results above with an extensive empirical study of several SSS and end-to-end algorithms on both the CIFAR-10 and ImageNet datasets.

1. We study several top-performing SSS architectures, and show that they all exhibit relatively small generalization gaps on both CIFAR-10 and ImageNet. We stress that we consider the case where the *same data* is used for both representation learning and classification, and hence it is by no means a-priori obvious that these algorithms should have small generalization gaps. See Figures 1 and 4 for sample results and Section 4 for more details.

2. We also show that the results of Zhang et al. (2017) do *not* replicate to SSS algorithms, in the sense that such algorithms, despite using an over-parameterized representation, are *not* able to fit random label noise.

3. We understake an empirical study of the robustness, rationality, and memorization gaps for both SSS and end-to-end supervised learning algorithms. We show that the robustness and rationality gaps are small for all these algorithms, while the memorization gap is small for SSS algorithms but can be large for end-to-end supervised learning. We show that the RRM bound is typically non-vacuous, and in fact, often close to tight, for a variety of SSS algorithms on the CIFAR-10 and ImageNet datasets, including SimCLR (which achieves test errors close to its supervised counterparts).

4. We demonstrate that replacing the memorization gap with the upper bound of Theorem II yields a *non-vacuous generalization bound* for a variety of SSS algorithms on CIFAR-10 and ImageNet. Moreover, this bound gets tighter with more data augmentation.

**Related Work.** There are many works on generalization bounds for supervised learning (e.g., Golowich et al. (2018); Neyshabur et al. (2017); Bartlett et al. (2017); Dziugaite & Roy (2017); Neyshabur et al. (2018); Cao & Gu (2019), and references therein). The related work section of

Arora et al. (2019) contains an extensive discussion of such bounds, and why more often than not the assumptions used do not hold in practice. Indeed, many such bounds give vacuous guarantees for modern architectures (such as the ones considered in this paper) that have the capacity to memorize their entire training set (Zhang et al., 2017). Some non-vacuous bounds are known; e.g., Zhou et al. (2019) gave a 96.5% bound on the error of MobileNet on ImageNet. Belkin et al. (2019); Nagarajan & Kolter (2019) showed some barriers for generalization gaps for standard end-to-end supervised learning. Similarly, standard approaches such as Rademacher complexity cannot directly bound SSS algorithms' generalization gap(see Remark 3.1).

Recently, Saunshi et al. (2019) and Lee et al. (2020) gave generalization bounds for self-supervised based classifiers. The two works considered special cases of SSS algorithms, such as *contrastive learning* and *pre-text tasks*. Both works make strong statistical assumptions of (exact or approximate) *conditional independence* relating the pre-training and classification tasks. For example, if the pre-training task is obtained by splitting a given image $x$ into two pieces $(x_1, x_2)$ and predicting $x_2$ from $x_1$, then Lee et al. (2020)'s results require $x_1$ and $x_2$ to be approximately independent conditioned on their class $y$. However, in many realistic cases, the two parts of the same image will share a significant amount of information not explained by the label. Our work applies to general SSS algorithms without such statistical assumptions, at the expense of assuming bounds on the robustness and rationality gaps. There have been works providing rigorous bounds on the robustness gap or related quantities (See Section 3.). However, as far as we know, the rationality gap has not been explicitly defined or studied before.

We provide a brief exposition of the various types of SSS methods in Section 4, and a more detailed discussion in Appendix D.1.

**Paper Organization.** Section 2 contains formal definitions and statements of our results. Section 3 provides an overview of prior work and our new results on the three gaps of the RRM bound. In Section 4, we describe our experimental setup and detail our empirical results. Section 5 concludes the paper and discusses important open questions. We defer proofs and additional experimental results to the appendix. Appendix B contains the proof of Theorem II, while Appendix C contains the proof of Theorem 3.2. Appendix D fully details our experimental setup.[1]

**Notation.** We use capital letters (e.g., $X$) for random variables, lower case letters (e.g., $x$) for a single value, and bold font (e.g., $\boldsymbol{x}$) for tuples (which will typically have dimension corresponding to the number of samples, denoted by $n$). We use $x_i$ for the $i$-th element of the tuple $\boldsymbol{x}$. We use calligraphic letters (e.g., $\mathcal{X}, \mathcal{D}$) for both sets and distributions.

## 2 FORMAL STATEMENT OF RESULTS

A *training procedure* is a (possibly randomized) algorithm $T$ that takes as input a train set $(\boldsymbol{x}, \boldsymbol{y}) = (x_i, y_i)_{i \in [n]} \in (\mathcal{X} \times \mathcal{Y})^n$ and outputs a classifier $f : \mathcal{X} \to \mathcal{Y}$. For our current discussion, we make no assumptions on the type of classifier output or the way that it is computed. We denote the distribution over training sets in $(\mathcal{X} \times \mathcal{Y})^n$ by $\mathcal{D}_{\text{train}}$ and the distribution over test samples in $\mathcal{X} \times \mathcal{Y}$ by $\mathcal{D}_{\text{test}}$.[2] The *generalization gap* of a training algorithm $T$ with respect to a distribution pair $\mathcal{D} = (\mathcal{D}_{\text{train}}, \mathcal{D}_{\text{test}})$ is the expected difference between its train accuracy (which we denote by $\text{Train}_{\mathcal{D},T}$) and its test performance (which we denote by $\text{Test}_{\mathcal{D},T}$). We will often drop subscripts such as $\mathcal{D}, T$ when they can be inferred from the context. We will also consider the *$\eta$-noisy experiment*, which involves computing the classifier $\tilde{f} = T(\boldsymbol{x}, \tilde{\boldsymbol{y}})$ where $\tilde{y}_i = y_i$ with probability $1 - \eta$ and is uniform over $\mathcal{Y}$ otherwise.

Our starting point is the following observation which we call the **RRM bound** (for **R**obustness, **R**ationality, and **M**emorization). The quantities appearing in it are defined in Table 1 and discussed more in depth in Section 3.

**Fact I** (*RRM bound*). *For every noise parameter $\eta > 0$, training procedure $T$ and distribution $\mathcal{D} = (\mathcal{D}_{\text{train}}, \mathcal{D}_{\text{test}})$ over training sets and test samples, the RRM bound with respect to $T$ and $\mathcal{D}$ is,*

---

[1] We provide our code and data in an anonymous repository on: http://github.com/ICLR2021-rep-gen/.

[2] The train and test data often stem from the same distribution (i.e., $\mathcal{D}_{\text{train}} = \mathcal{D}_{\text{test}}^n$), but not always (e.g., it does not hold if we use data augmentation). $\mathcal{D}_{\text{test}}$ enters the RRM bound only via the rationality gap, so the assumption of small rationality may be affected if $\mathcal{D}_{\text{train}} \neq \mathcal{D}_{\text{test}}^n$, but the RRM bound still holds.

$$\underbrace{\mathsf{Train} - \mathsf{Test}}_{\substack{\textit{Generalization} \\ \textit{gap}}} \leq \underbrace{\left[\mathsf{Train} - \mathsf{Train}(\eta)\right]_+}_{\substack{\textit{Robustness} \\ \textit{gap}}} + \underbrace{\left[\mathsf{NTrain}(\eta) - \mathsf{Test}\right]_+}_{\substack{\textit{Rationality} \\ \textit{gap}}} + \underbrace{\left[\mathsf{Train}(\eta) - \mathsf{NTrain}(\eta)\right]_+}_{\substack{\textit{Memorization} \\ \textit{gap}}}$$

*where we denote* $x_+ = \max(x, 0)$.

**Table 1** – The measurements of accuracy in the RRM bound, all with respect to a training algorithm $T$, distributions $(\mathcal{D}_{\text{train}}, \mathcal{D}_{\text{test}})$ and parameter $\eta > 0$. The *robustness gap* is $\max(\mathsf{Train} - \mathsf{Train}(\eta), 0)$, the *rationality gap* is $\max(\mathsf{NTrain}(\eta) - \mathsf{Test}, 0)$, and the *memorization gap* is $\max(\mathsf{Train}(\eta) - \mathsf{NTrain}(\eta), 0)$.

| Quantity | Training | Measurement |
|---|---|---|
| $\mathsf{Test}_{\mathcal{D},T}$ | $f = T(\boldsymbol{x}, \boldsymbol{y})$ for $(\boldsymbol{x}, \boldsymbol{y}) \sim \mathcal{D}_{\text{train}}$ | $\mathbf{Pr}[f(x) = y]$ for $(x, y) \sim \mathcal{D}_{\text{test}}$. |
| $\mathsf{Train}_{\mathcal{D},T}$ | $f = T(\boldsymbol{x}, \boldsymbol{y})$ for $(\boldsymbol{x}, \boldsymbol{y}) \sim \mathcal{D}_{\text{train}}$ | $\mathbf{Pr}[f(x_i) = y_i]$ for train sample $(x_i, y_i)$. |
| $\mathsf{Train}_{\mathcal{D},T}(\eta)$ | $\tilde{f} = T(\boldsymbol{x}, \tilde{\boldsymbol{y}})$ for $(\boldsymbol{x}, \boldsymbol{y}) \sim \mathcal{D}_{\text{train}}$, $\tilde{y}_i = y_i$ w.p. $1 - \eta$, uniform o/w | $\mathbf{Pr}[\tilde{f}(x_i) = y_i]$ for train sample $(x_i, \tilde{y}_i)$ where $y_i$ *original* label for $x_i$. |
| $\mathsf{NTrain}_{\mathcal{D},T}(\eta)$ | $\tilde{f} = T(\boldsymbol{x}, \tilde{\boldsymbol{y}})$ for $(\boldsymbol{x}, \boldsymbol{y}) \sim \mathcal{D}_{\text{train}}$, $\tilde{y}_i = y_i$ w.p. $1 - \eta$, uniform o/w | $\mathbf{Pr}[\tilde{f}(x_i) = y_i | \tilde{y}_i \neq y_i]$ for a corrupted train sample $x_i$ where $y_i$ *original* label for $x_i$. |

The RRM bound is but an observation, as it directly follows from the fact that $x_+ \geq x$ for every $x$. However, it is a very useful one. As mentioned above, for natural algorithms, we expect both the *robustness* and *rationality* components of this gap to be small, and hence the most significant component is the *memorization gap*. Our main theoretical result is a bound on this gap:

**Theorem II** (*Memorization gap bound*). *Let* $T = (T_{pre}, T_{fit})$ *be an SSS training procedure obtained by first training* $T_{pre}$ *on* $\boldsymbol{x} \in \mathcal{X}^n$ *to get a representation* $r : \mathcal{X} \to \mathcal{R}$ *and then training* $T_{fit}$ *on* $(r(\boldsymbol{x}), \boldsymbol{y})$ *for* $\boldsymbol{y} \in \mathcal{Y}^n$ *to obtain a classifier* $g : \mathcal{R} \to \mathcal{Y}$, *with the final classifier* $f : \mathcal{X} \to \mathcal{Y}$ *defined as* $f(x) = g(r(x))$. *Then, for every noise parameter* $\eta > 0$ *and distribution* $\mathcal{D}$ *over* $\mathcal{X}^n \times \mathcal{Y}^n$:

$$\textit{Memorization gap}(T) = \mathsf{Train}_{T,\mathcal{D}}(\eta) - \mathsf{NTrain}_{T,\mathcal{D}}(\eta) \leq O\left(\sqrt{\frac{\mathsf{C}_\eta(T_{fit})}{n} \cdot \frac{1}{\eta}}\right)$$

*where* $\mathsf{C}_\eta(T_{fit})$ *is a complexity measure of the second phase training procedure, which in particular is upper bounded by the number of bits required to describe the classifier* $g$ *(See Definition 2.3.).*

### 2.1 COMPLEXITY MEASURES

We now define three complexity measures, all of which can be plugged in as the measure in Theorem II. The first one, $\mathsf{C}^{\text{mdl}}$, is the minimum description length of a classifier in bits. At a first reading, the reader can feel free to skip the description of the other two measures $\mathsf{C}^{\text{pc}}$ and $\mathsf{C}^{\text{dc}}$. These are superficially similar to Rademacher Complexity (cf. Bartlett & Mendelson (2002)) in the sense that they capture the ability of the hypothesis to correlate with random noise but crucially depend on the algorithm used rather than the class of concepts (see Remark 3.1).

**Definition 2.3** (*Complexity of training procedures*). Let $T$ be a training procedure taking as input a set $(\boldsymbol{r}, \boldsymbol{y}) = \{(r_i, y_i)\}_{i=1}^n \in (\mathcal{R} \times \mathcal{Y})^n$ and outputting a classifier $g : \boldsymbol{r} \to \mathcal{Y}$ and let $\eta > 0$. For every training set $(\boldsymbol{r}, \boldsymbol{y})$, we define the following three complexity measures with respect to $\boldsymbol{r}, \boldsymbol{y}, \eta$:

- The *minimum description length* of $T$ is defined as $\mathsf{C}^{\text{mdl}}_{\boldsymbol{r}, \boldsymbol{y}, \eta}(T) := H(g)$ where we consider the model $g$ as a random variable arising in the $\eta$-noisy experiment.[3]

- The *prediction complexity* of $T$ is defined as $\mathsf{C}^{\text{pc}}_{\boldsymbol{r}, \boldsymbol{y}, \eta}(T) := \sum_{i=1}^n I(g(r_i); \tilde{y}_i)$ where the $\tilde{y}_i$'s are the labels obtained in the $\eta$-noisy experiment.

- The (unconditional) *deviation complexity* of $T$ is defined as $\mathsf{C}^{\text{dc}}_{\boldsymbol{r}, \boldsymbol{y}, \eta}(T) := n \cdot I(g(r_i) - y_i ; \tilde{y}_i - y_i)$ where the random variables above are taken over $i \sim [n]$ and subtraction is done modulo $|\mathcal{Y}|$, identifying $\mathcal{Y}$ with the set $\{0, \ldots, |\mathcal{Y}| - 1\}$.

---

[3]The name "minimum description length" is justified by the operational definition of entropy relating it to the minimum amortized length of a prefix-free encoding of a random variable.

Conditioned on $\boldsymbol{y}$ and the choice of the index $i$, the deviations $g(r_i) - y_i$ and $\tilde{y}_i - y_i$ determine the predictions $g(r_i)$ and noisy labels $\tilde{y}_i$, and vice versa. Hence we can think of $\mathsf{C}^{\text{dc}}$ as an "averaged" variant of $\mathsf{C}^{\text{pc}}$, where we make the choice of the index $i$ part of the sample space for the random variables. While we expect the two measures to be approximately close, the fact that $\mathsf{C}^{\text{dc}}$ takes $i$ into the sample space makes it easier to estimate this quantity in practice without using a large number of executions (See Figure D.2 for convergence rates.). The measure $\mathsf{C}^{\text{mdl}}$ is harder to evaluate in practice, as it requires finding the optimal compression scheme for the classifier. Appendix B contains the full proof of Theorem II. It is obtained by showing that: **(i)** for every $\boldsymbol{r}, \boldsymbol{y}, \eta$, and $T$ it holds that $\mathsf{C}^{\text{dc}}_{\boldsymbol{r},\boldsymbol{y},\eta}(T) \leq \mathsf{C}^{\text{pc}}_{\boldsymbol{r},\boldsymbol{y},\eta}(T) \leq \mathsf{C}^{\text{mdl}}_{\boldsymbol{r},\boldsymbol{y},\eta}(T)$, and **(ii)** for every SSS algorithm $T = (T_{\text{pre}}, T_{\text{fit}})$ and distribution $\mathcal{D} = (\mathcal{D}_{\text{train}}, \mathcal{D}_{\text{test}})$, the memorization gap of $T$ is at most

$$\sqrt{\mathsf{C}^{\text{dc}}_{T_{\text{pre}}(\boldsymbol{x}),\boldsymbol{y},\eta}(T_{\text{fit}})} \Big/ \left(\eta\sqrt{2n}\right) . \tag{1}$$

It is the quantity (1) that we compute in our experiments.

## 3 The three gaps

We now briefly describe what is known and what we prove about the three components of the RRM bound. We provide some additional discussions in Appendix E, including "counter-examples" of algorithms that exhibit large values for each one of these gaps.

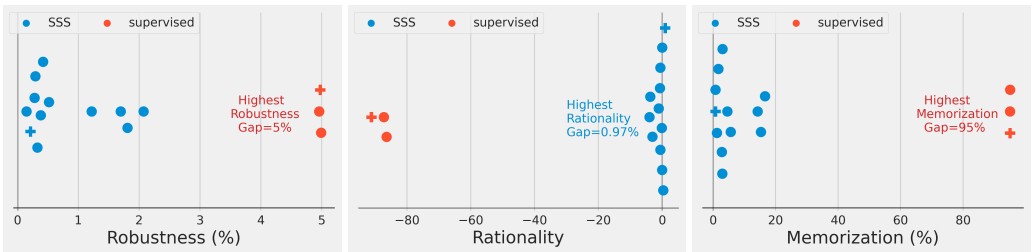

**Figure 2 – Robustness, Rationality, and Memorization for CIFAR-10** Each blue point is a different combination of (architecture + self-supervised task + fitting algorithm). Each red point is a different architecture trained end-to-end with supervision. We use the '+' marker to denote the two best models of each type (SSS and supervised). No augmentations were added. Noise is $5\%$. Details in Appendix D.3

**The robustness gap.** The robustness gap measures the decrease in training accuracy from adding $\eta$ noisy labels, measured with respect to the clean labels. The robustness gap and related notions such as *noise stability or tolerance* have been studied in various works (cf. Frénay & Verleysen (2013); Manwani & Sastry (2013)). *Interpolating classifiers* (with zero train error) satisfy $\mathsf{Train}(\eta) \geq 1 - \eta$ and hence their robustness gap is at most $\eta$ (See left panel of Figure 2). In SSS algorithms, since the representation is learned without using labels, the injection of label noise only affects the simple classifier, which is often *linear*. Robustness guarantees for linear classifiers have been given previously by Rudin (2005). While proving robustness bounds is not the focus of this paper, we note in the appendix some simple bounds for least-squares minimization of linear classifiers and the (potentially inefficient) Empirical Risk Minimization algorithm (see Appendices F and G). Empirically, we observe that the robustness gap of SSS algorithms is often significantly smaller than $\eta$. (See left panels of Figure 2 and Figure 3.)

**The memorization gap.** The *memorization gap* corresponds to the algorithm's ability to fit the noise (i.e., the gap increases with the number of fit noisy labels). If, for example, the classifier output is *interpolating*, i.e., it satisfies $f(x_i) = \tilde{y}_i$ for every $i$, then accuracy over the noisy samples will be 0 (since for them $y_i \neq \tilde{y}_i$). In contrast, the overall accuracy will be in expectation at least $1 - \eta$ which means that the memorization gap will be $\approx 1$ for small $\eta$. However, we show empirically (see right panels of Figures 2 and 3) that the memorization gap is small for many SSS algorithms and *prove* a bound on it in Theorem II. When combined with small rationality and robustness, this bound results in non-vacuous generalization bounds for various real settings (e.g., 48% for ResNet101 with SimCLRv2 on ImageNet, and as low as 4% for MoCo V2 with ResNet-18 on CIFAR-10). Moreover, unlike other generalization bounds, our bound decreases with *data augmentation* (See Figure 5.).

**Remark 3.1** (*Memorization vs. Rademacher complexity*)**.** The memorization gap, as well the complexity measures defined in Section 2.1 have a superficial similarity to *Rademacher complexity* (Bartlett & Mendelson, 2002), in the sense that they quantify the ability of the output classifier to fit noise. One difference is that Rademacher complexity is defined with respect to $100\%$ noise, while we consider the $\eta$-noisy experiment for small $\eta$. A more fundamental difference is that Rademacher complexity is defined via a supremum over all classifiers in some class. The final classifiers of SSS algorithms are obtained by a composition of the complex representation and simple classifier. This composed classifier will in general have high Radamacher complexity, and in particular we would not be able to prove non-vacuous bounds on it using Radamacher complexity. We cannot ignore the complexity of the representation in Radamacher-complexity based analysis of SSS algorithms since the representation is learned *using the same data* that is later used for classification. In fact, there are examples of SSS algorithms with simple classifiers that have large generalization gaps (see Section 3.1). This shows that Radamacher complexity bounds for the class of simple classifiers cannot, on their own, be used to derive generalization bounds.

Zhang et al. (2017) demonstrated a lower bound on the Radamacher complexity of modern deep networks, by showing that modern end-to-end supervised learning algorithm can fit $100\%$ of their label noise. Our experiments show that this is *not the case* for SSS algorithms, which can only fit $15\%$-$25\%$ of the CIFAR-10 training set when the labels are completely random (See Table D.1 in the appendix.). However, absence of evidence is not evidence of absence, and the fact that empirically SSS algorithms do not fit the noise, does not imply that the Radamacher complexity of the resulting class is small, nor does it, by its own, automatically imply a small generalization gap.

## 3.1 THE RATIONALITY GAP

Unlike the other quantities defined above, the *rationality gap* is novel and less intuitive, and so we discuss it more in depth. The rationality gap, like all other quantities in the RRM bound, applies to *any* learning procedure and not only to SSS algorithms. Indeed, our empirical results show that rationality is typically small for both SSS and end-to-end algorithms, and so it is not this gap but rather the *memorization gap* that accounts for the difference in their generalization behavior.

To build intuition for the rationality gap, consider an example of a training procedure $T$ that on input a train set $S$, has $70\%$ test accuracy and a $10\%$ rationality gap with noise parameter $\eta = 5\%$. In the $\eta$-noisy experiment, the classifier $\tilde{f}$ output by $T$ recovers the *original uncorrupted label* for $80\%$ of the $\approx \eta \cdot n$ datapoints for which it received the *wrong* labels. In contrast, $10\%$ rationality gap means the same classifier will only succeed in recovering the label of $70\%$ of unseen test samples.

Intuitively, such a classifier is being "irrational" or "inconsistent" in the sense that it succeeds better on datapoints on which it was given the *wrong* label, then on datapoints on which it was given *no label at all*. (In error-correcting code parlance, it handles *corruption* errors better than *erasure* errors.) We can turn this intuition into a formal argument, by giving a transformation from such a training algorithm $T$ to an algorithm $T'$ that achieves roughly $80\%$ test accuracy. On input a fresh unseen datapoint $x$, the algorithm $T'$ chooses a *random* label $\tilde{y} \sim \mathcal{Y}$, runs $T$ on the train set $S \cup \{(x, \tilde{y})\}$ to obtain some classifier $\tilde{f}$, and outputs $\tilde{f}(x)$. Up to low-order terms, $T'$ will achieve test accuracy at least as good as the performance of $T$ on noisy datapoints, which is $80\%$. The above reasoning leads to the proof of the following theorem (see also Appendix C):

**Theorem 3.2** (*Performance on the table theorem, informal*)**.** *For every training procedure $T$ and distribution $\mathcal{D}_{test}$, $\mathcal{D}_{train} = \mathcal{D}_{test}^n$, there exists a training procedure $T'$ satisfying $\mathsf{Test}_{T'} \geq \mathsf{Test}_T + $ rationality gap$(T) - $ robustness gap$(T) - o(1)$.*

**Why do natural algorithms have a small rationality gap?** Empirically, the rationality gap is often small or zero for both SSS and end-to-end supervised learning algorithms, particularly for better-performing ones. (See middle panels of Figure 2 and Figure 3.) Theorem 3.2 provides an "economic explanation" for this phenomenon: a rational agent would not use a classifier with a positive rationality gap since this amounts to "leaving performance on the table". However, this transformation comes at a high computational cost; inference for the classifier produced by $T'$ is as expensive as retraining from scratch. Hence Theorem 3.2 does not fully explain why natural algorithms tend to have small rationality gap. In this paper we take low rationality gap as an empirically-justifiable assumption. We believe that both proving that natural algorithms have small rationality gaps, as well

as coming up with computationally efficient transformations to extract performance from rationality gaps, are important open questions.

**Does assuming small rationality gap trivialize generalization?**   Since the definition of the rationality gap involves the test accuracy, the reader might wonder if assuming small rationality is not tantamount to assuming a small generalization gap. However, there is nothing "irrational" about a large generalization gap, and indeed many excellent classifiers have 100% train accuracy. In contrast, it *is* irrational to "leave performance on the table" and use a classifier with test accuracy $p$ when it can be transformed into one with significantly better accuracy. Concretely, our empirical studies show that the rationality gap is uniformly small, even for end-to-end classifiers that have large generalization gaps. Hence, by itself, rationality is not enough to guarantee small generalization gap.

**Is assuming small rationality gap even needed?**   Since SSS algorithms use simple classifiers, the reader may wonder why we need the small-rationality gap assumption and cannot directly prove generalization bounds using standard tools such as Rademacher complexity. The issue is that the representation used by SSS algorithms is still sufficiently over-parameterized to allow memorizing the training set. As a pedagogical example, consider a representation-learning procedure that maps a label-free training set $x$ to a representation $r : \mathcal{X} \to \mathcal{R}$ under which the differently labeled $x$'s are linearly separable. Moreover, suppose that the representation space has dimension much smaller than $n$, and hence a linear classifier would have small complexity under any reasonable measure. Without access to the labels, we can transform $r$ to a representation $r'$ that on input $x$ outputs $r(x)$ if $x$ is in the training set, and outputs the all-zero vector (or another trivial value) otherwise. Given sufficiently many parameters, the representation $r'$ (or a close-enough approximation) can be implemented by a neural network. Since $r$ and $r'$ are identical on the training set, a learning procedure using $r'$ will have the same train accuracy and (small) memorization gap. However, the generalization gap of such a procedure will be large, since it will not achieve better than trivial accuracy on unseen test examples. The issue here is *not* that the representation "memorizes" the train set. Representations of practical SSS algorithms are highly over-parameterized and are quite likely to memorize specific aspects of the training set. Rather, the issue is that the representation artificially behaves differently on test points in a way that decreases its performance. It is the latter property that makes the classifier "irrational", and violates the small rationality gap assumption.

## 4    EMPIRICAL STUDY OF THE RRM BOUND

In support of our theoretical results, we conduct an extensive empirical study of the three gaps and empirically evaluate the bound from Equation (1) for a variety of SSS algorithms for the CIFAR-10 and ImageNet datasets. We provide a summary of our setup and findings below. For a full description of the algorithms and hyperparameters, see Appendix D.

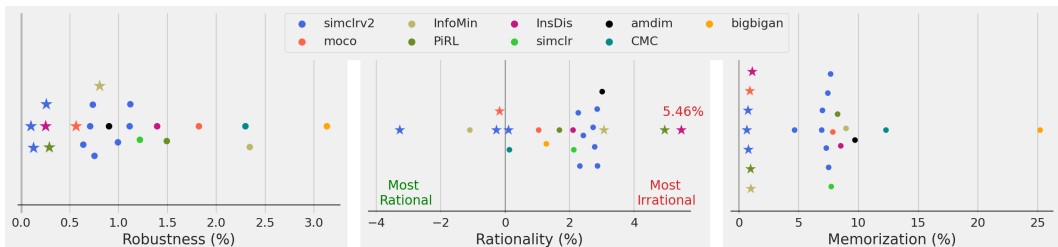

**Figure 3 – Robustness, Rationality and Memorization for ImageNet.** Each point represents a different combination of self-supervised learning algorithm (e.g., SimCLR), backbone architecture (e.g., ResNet-50) and simple classifier (e.g., linear classification). Star indicates experiments with 10 augmentations per training sample. Noise level is $\eta = 5\%$. Full experimental details in Section D.

**SSS Algorithms** ($T_{\text{pre}}, T_{\text{fit}}$).   We consider various self-supervised training algorithms that learn a representation *without* explicit training labels. In our study, we include methods based on contrastive learning such as Instance Discrimination (Wu et al., 2018), MoCoV2 (He et al., 2020), SimCLR (Chen et al., 2020a;b), AMDIM (Bachman et al., 2019), CMC (Tian et al., 2019), InfoMin (Tian et al., 2020) as well as adversarial methods such as BigBiGAN (Donahue & Simonyan, 2019). For the second phase of training (also known as the evaluation phase (Goyal et al., 2019)), we consider

simple models such as regularized linear regression, or small Multi-Layer Perceptrons (MLPs). For each evaluation method, we run two experiments: 1) the clean experiment where we train $T_{\text{fit}}$ on the data and labels $(\boldsymbol{x}, \boldsymbol{y})$; 2) the $\eta$-noisy experiment where we train $T_{\text{fit}}$ on $(\boldsymbol{x}, \tilde{\boldsymbol{y}})$ where $\tilde{\boldsymbol{y}}$ are the $\eta$ noised labels. Unless specified otherwise we set the noise to $\eta = 5\%$.

**Adding augmentations.** We investigate the effect of data augmentation on the three gaps and the theoretical bound. For each training point, we sample $t$ random augmentations ($t = 10$ unless stated otherwise) and add it to the train set. Note that in the noisy experiment two augmented samples of the same original point might be assigned with different labels. We use the same augmentation used in the corresponding self-supervised training phase.

**Results.** Figures 1 and 2 provide a summary of our experimental results for CIFAR-10. The robustness and rationality gaps are close to zero for most SSS algorithms, while the memorization gap is usually the dominant term, especially so for models with larger generalization gap. Moreover, we see that $C^{\text{dc}}$ often produces a reasonably tight bound for the memorization gap, leading to a generalization bound that can be as low as $5$-$10\%$. In Figures 3 and 4 we give a summary of our experimental results for SSS algorithms on ImageNet. Again, the rationality and robustness gaps are bounded by small constants. Notice, that adding augmentations reduces memorization, but may lead to an increase in the rationality gap. This is also demonstrated in Figure 5 where we vary the number of data augmentations systematically for one SSS algorithm (AMDIM) on CIFAR-10. Since computing the Theorem II bound for ImageNet is computationally expensive (See Appendix D.5.1.) we compute it only for two algorithms, which achieve a non-vacuous generalization bound of $48\%$.

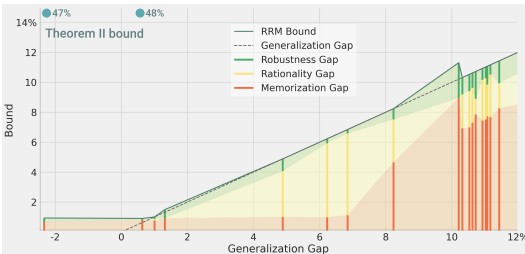

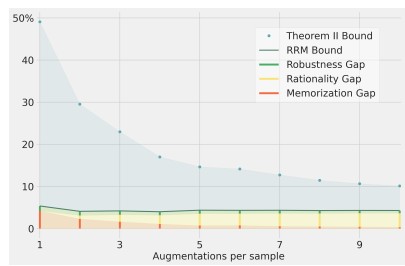

**Figure 4** – The RRM bound of SSS methods on ImageNet, with models sorted by the generalization gap. We plot the robustness, rationality and memorization gaps. Similar to Figure 1, for most models, the bound is tight and is dominated by the memorization gap. Theorem II bound is marked for the two leftmost models (we did not evaluate it for the others, for computational reasons).

**Figure 5** – Empirical RRM for the AMDIM SSS model on CIFAR-10 with increasing number of augmentations. While robustness and memorization gaps decrease, and so does our generalization bound, the rationality gap increases since $\mathcal{D}_{\text{train}}$ and $\mathcal{D}_{\text{test}}$ grow apart.

## 5 CONCLUSIONS AND OPEN QUESTIONS

This work demonstrates that SSS algorithms have small generalization gaps. While our focus is on the *memorization gap*, our work motivates more investigation of both the *robustness* and *rationality* gaps. In particular, we are not aware of any rigorous bounds for the rationality gap of SSS algorithms, but we view our "performance on the table" theorem (Theorem 3.2) as a strong indication that it is close to zero for natural algorithms. Given our empirical studies, we believe the assumptions of small robustness and rationality conform well to practice.

Our numerical bounds are still far from tight, especially for ImageNet, where evaluating the bound (more so with augmentations) is computationally expensive. Nevertheless, we find it striking that already in this initial work, we get non-vacuous (and sometimes quite good) bounds. Furthermore, the fact that the empirical RRM bound is often close to the generalization gap, shows that there is significant room for improvement.

Overall, this work can be viewed as additional evidence for the advantages of SSS algorithms over end-to-end supervised learning. Moreover, some (very preliminary) evidence shows that end-to-end supervised learning implicitly separates into a representation learning and classification phases (Morcos et al., 2018). Understanding the extent that supervised learning algorithms implicitly perform SSS learning is an important research direction in its own right. To the extent this holds, our work might shed light on such algorithms' generalization performance as well.

## 6 ACKNOWLEDGEMENTS

We thank Dimitris Kalimeris, Preetum Nakkiran, and Eran Malach for comments on early drafts of this work. This work supported in part by NSF award CCF 1565264, IIS 1409097, DARPA grant W911NF2010021, and a Simons Investigator Fellowship. We also thank Oracle and Microsoft for grants used for computational resources. Y.B is partially supported by MIT-IBM Watson AI Lab. Work partially performed while G.K. was an intern at Google Research.

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

## A    MUTUAL INFORMATION FACTS

**Lemma A.1 .** *If $A, B$ are two Bernoulli random variables with nonzero expectation then*

$$| \mathbb{E}[A|B=1] - \mathbb{E}[A]| \leq \sqrt{\tfrac{1}{2}I(A;B)}/\mathbb{E}[B]$$

*Proof.* A standard relation between mutual information and KL-divergence gives

$$I(A;B) = D_{KL}(p_{A,B}||p_A p_B).$$

On the other hand, by the Pinsker inequality,

$$\sup_{S \subseteq \{0,1\} \times \{0,1\}} |p_{A,B}(S) - p_{A \times B}(S)| \leq \sqrt{\frac{1}{2}D_{KL}(p_{A,B}||p_A p_B)} = \sqrt{\frac{1}{2}I(A,B)}.$$

Thus (letting $S = \{(1,1)\}$),

$$|\mathbf{Pr}[A=1, B=1] - \mathbf{Pr}[A=1]\mathbf{Pr}[B=1]| \leq \sqrt{\tfrac{1}{2}I(A,B)}.$$

Consequently,

$$|\mathbb{E}[A|B=1] - \mathbb{E}[A]| \leq \sqrt{\tfrac{1}{2}I(A,B))}/\mathbb{E}(B)$$

$\square$

**Lemma A.2 .** *For three random variables $W, X, Y$, s.t. $X$ and $Y$ are independent,*

$$I(W;X,Y) \geq I(W;X) + I(W;Y)$$

*Proof.* Using the chain rule for mutual information we have:

$$I(W;X,Y) = I(W;X) + I(W;Y|X)$$

Since $X, Y$ are independent, $H(Y|X) = H(Y)$ and since conditioning only reduces entropy, we have $H(Y|W,X) \leq H(Y|W)$. Combining the two we get,

$$\begin{aligned} I(W;Y|X) &= H(Y|X) - H(Y|W,X) \\ &\geq H(Y) - H(Y|W) \\ &= I(W;Y) \end{aligned}$$

Thus we have that $I(W;X,Y) \geq I(W;X) + I(W;Y)$. $\square$

Note that by induction we can extend this argument to show that $I(W;X_1, ..., X_n) \geq \sum I(W;X_i)$ where $X_i$ are mutually independent.

## B    SIMPLE CLASSIFIERS IMPLY SMALL MEMORIZATION GAP

In this appendix we we prove our main theoretical result (Theorem B.4). We will start by giving a formal definition of SSS algorithms and restating the definition of our complexity measures.

**Definition B.1** (*SSS Algorithms, restated*)**.** An *SSS algorithm* over $(\mathcal{X} \times \mathcal{Y})^n$ is a procedure $T = (T_{\text{pre}}, T_{\text{fit}})$ that takes as input a set $(\boldsymbol{x}, \boldsymbol{y})$ and operates as follows:

1. $T_{\text{pre}}$ takes the (label free) data points $\boldsymbol{x}$ as input and outputs a *representation* $r : \mathcal{X} \to \mathcal{R}$ for some set $\mathcal{R}$;

2. On input the points $\{(r(x_i), y_i)\}_{i=1}^n$, $T_{\text{fit}}$ outputs a *simple classifier* $g : \mathcal{R} :\to \mathcal{Y}$;

3. The output is a classifier $f : \mathcal{X} \to \mathcal{Y}$ defined as $f(x) = g(r(x))$ for every $x \in \mathcal{X}$.

We now restate the definitions of our complexity measure.

**Definition B.2** (*Complexity of training procedures, restated*)**.** Let $T$ be a training procedure taking as input $(\boldsymbol{r}, \boldsymbol{y}) = \{(r_i, y_i)\}_{i=1}^n \in (\mathcal{R} \times \mathcal{Y})^n$ and outputting a classifier $g : \boldsymbol{r} \to \mathcal{Y}$ and let $\eta > 0$. For every training set $(\boldsymbol{r}, \boldsymbol{y})$:

- The *minimum description length* of $T$ with respect to $\boldsymbol{r}, \boldsymbol{y}, \eta$ is defined as $\mathsf{C}^{\mathrm{mdl}}_{\boldsymbol{r}, \boldsymbol{y}, \eta}(T) = H(g)$ where $g$ is the random variable $T(\boldsymbol{r}, \tilde{\boldsymbol{y}})$ in the $\eta$ noisy experiment.

- The *prediction complexity* of $T$ with respect to $\boldsymbol{r}, \boldsymbol{y}, \eta$ is defined as,

$$\mathsf{C}^{\mathrm{pc}}_{\boldsymbol{r}, \boldsymbol{y}, \eta}(T) := \sum_{i=1}^n I(g(r_i); \tilde{y}_i)$$

where $g(r_i)$ and $\tilde{y}_i$ are viewed as random variables over the sample space induced by choosing $\tilde{\boldsymbol{y}}$ according to the $\eta$-noisy experiment w.r.t. $\boldsymbol{y}$ and letting $g = T(\boldsymbol{x}, \tilde{\boldsymbol{y}})$.

- The *deviation complexity of $T$ with respect to $\boldsymbol{r}, \boldsymbol{y}, \eta$* is defined as

$$\mathsf{C}^{\mathrm{dc}}_{\boldsymbol{r}, \boldsymbol{y}, \eta}(T) := n \cdot I(\Delta; N)$$

where $\Delta = g(r_i) - y_i \pmod{|\mathcal{Y}|}$ and $N = \tilde{y}_i - y_i \pmod{|\mathcal{Y}|}$ are random variables taken over both the above sample space **and the choice of** $i \sim [n]$ and identifying $\mathcal{Y}$ with $\{0, \ldots, |\mathcal{Y}| - 1\}$.

The following theorem shows that $\mathsf{C}^{\mathrm{dc}}$ is upper bounded by $\mathsf{C}^{\mathrm{pc}}$, which in turn is bounded by the operational entropy of $g$.

**Theorem B.3** (*Relation of complexity measures*)**.** *For every $\boldsymbol{r}, \boldsymbol{y}, \eta > 0$, and $T$*

$$\mathsf{C}^{\mathit{dc}}_{\boldsymbol{r}, \boldsymbol{y}, \eta}(T) \leq \mathsf{C}^{\mathit{pc}}_{\boldsymbol{r}, \boldsymbol{y}, \eta}(T) \leq \mathsf{C}^{\mathit{mdl}}(T)$$

*where $g$ is the classifier output by $T$ (considered as a random variable).*

*Proof.* Fix $T, \boldsymbol{r}, \boldsymbol{y}, \eta$. We get $\tilde{\boldsymbol{y}}$ by choosing i.i.d random variables $N_1, \ldots, N_n$, each equalling $0$ with probability $1 - \eta$ and uniform otherwise, and letting $\tilde{y}_i = y_i + N_i \pmod{|\mathcal{Y}|}$.

We start by proving the second inequality $\mathsf{C}^{\mathrm{pc}}_{\boldsymbol{r}, \boldsymbol{y}, \eta}(T) \leq H(g)$. Let $g = T(\boldsymbol{r}, \tilde{\boldsymbol{y}})$ and define $\boldsymbol{p} = (g(r_1), \ldots, g(r_n))$ be the vector of predictions. Then,

$$\mathsf{C}^{\mathrm{pc}}_{\boldsymbol{r}, \boldsymbol{y}, \eta}(T) = \sum_i I(p_i; \tilde{y}_i) = \sum_i I(p_i; N_i) \tag{2}$$

with the last equality holding since for fixed $y_i$, $N_i$ determines $\tilde{y}_i$ and vice versa. However, since the full vector $\boldsymbol{p}$ contains only more information than $p_i$, the right-hand side of (2) is at most $\sum_{i=1}^n I(\boldsymbol{p}; N_i) \leq I(\boldsymbol{p}; N_1, \ldots, N_n)$, using the fact that $N_i$ random variables are independent (see Lemma A.2). For a fixed $\boldsymbol{r}$, the value of $\boldsymbol{p}$ is completely determined by $g$ and hence the entropy of $\boldsymbol{p}$ is at most $H(g)$, establishing the second inequality of the theorem.

We now turn to the first inequality $\mathsf{C}^{\mathrm{dc}}_{\boldsymbol{r}, \boldsymbol{y}, \eta}(T) \leq \mathsf{C}^{\mathrm{pc}}_{\boldsymbol{r}, \boldsymbol{y}, \eta}(T)$. Let $\Delta_i = p_i - y_i \pmod{|\mathcal{Y}|}$. Then,

$$\tfrac{1}{n} \mathsf{C}^{\mathrm{pc}}_{\boldsymbol{r}, \boldsymbol{y}, \eta}(T) = \mathop{\mathbb{E}}_{j \sim [n]} I(p_j; N_j) = \mathop{\mathbb{E}}_{j \sim [n]} I(\Delta_j; N_j) \tag{3}$$

since $p_i$ determines $\Delta_i$ and vice versa. But, since $N_j = N | i = j$ and $\Delta_j = \Delta | i = j$ (where $N, \Delta$ are the random variables defined in Definition B.2), the right-hand side of (3) equals

$$\mathop{\mathbb{E}}_{j \sim [n]} I(\Delta; N | i = j) = \mathop{\mathbb{E}}_{j \sim [n]} H(N | i = j) - H(N | \Delta, i = j) . \tag{4}$$

Since $N_1, \ldots, N_n$ are identically distributed, $H(N | i = j) = H(N)$ which means that the right-hand side of (4) equals

$$H(N) - \mathop{\mathbb{E}}_{j \sim [n]} H(N | \Delta, i = j) \geq H(N) - H(N | \Delta) = I(\Delta; N)$$

with the inequality holding since on average conditioning reduces entropy. By definition $I(\Delta; N) = \frac{1}{n} \mathsf{C}^{\mathrm{dc}}_{\boldsymbol{r}, \boldsymbol{y}, \eta}(T)$, establishing what we wanted to prove. $\qquad \square$

The complexity measures $C^{pc}$ and $C^{dc}$ are defined with respect to a *fixed* train set $(\boldsymbol{r}, \boldsymbol{y})$, rendering them applicable for single training sets such as CIFAR-10 and ImageNet that arise in practice. If $\mathcal{D}$ is a distribution over $(\boldsymbol{r}, \boldsymbol{y})$, then we define the complexity measures $C^{pc}$ and $C^{dc}$ with respect to $\mathcal{D}$ as the average of the corresponding measure with respect to $(\boldsymbol{r}, \boldsymbol{y}) \sim \mathcal{D}$. We now restate Theorem II:

**Theorem B.4** (*Theorem II, restated*). *Let $T = (T_{pre}, T_{fit})$ be a training procedure obtained by first training $T_{pre}$ on $\boldsymbol{x} \in \mathcal{X}^n$ to obtain a representation $r : \mathcal{X} \to \mathcal{R}$ and then training $T_{fit}$ on $(r(\boldsymbol{x}), \boldsymbol{y}))$ where $\boldsymbol{y} \in \mathcal{Y}^n$ to obtain a classifier $g : \mathcal{R} \to \mathcal{Y}$. Then, for every noise parameter $\eta > 0$ and distribution $\mathcal{D}_{train}$ over $(\mathcal{X}, \mathcal{Y})^n$,*

$$\textit{Memorization gap}(T) = \mathsf{Train}_{\mathcal{D}_{train},T}(\eta) - \mathsf{NTrain}_{\mathcal{D}_{train},T}(\eta) \leq \sqrt{\frac{C^{dc}_{\mathcal{D}_r,\eta}(T_{fit})}{2n}} \cdot \frac{1}{\eta}$$

*where $\mathcal{D}_r$ is the distribution over $(\mathcal{R} \times \mathcal{Y})^n$ induced by $T_{pre}$ on $\mathcal{D}_{train}$.*

Note that the bound on the right-hand side is expressed only in terms of the complexity of the second stage $T_{\text{fit}}$ and is independent of the complexity of $T_{\text{pre}}$. The crux of the proof is showing (close to) independence between the corrupted indices and prediction deviation of $g$ resulting from the noise.

*Proof.* Let $(\boldsymbol{r}, \boldsymbol{y})$ be sampled by first drawing $(\boldsymbol{x}, \boldsymbol{y}) \sim \mathcal{D}_{\text{train}}$ over $(\mathcal{X} \times \mathcal{Y})^n$ then applying $\boldsymbol{r} = r(\boldsymbol{x})$ where $r = T_{\text{pre}}(\boldsymbol{x})$. Consider the sample space of sampling $\tilde{\boldsymbol{y}}$ according to the $\eta$-noisy distribution with respect to $Y$, computing $g = T_{\text{fit}}(\boldsymbol{r}, \tilde{\boldsymbol{y}})$, and sampling $i \sim [n]$. We define the following two Bernoulli random variables over this sample space:

$$Z = \mathbb{1}_{\Delta=0} = \begin{cases} 1 & g(R_i) = y_i \\ 0 & otherwise \end{cases} ; \qquad B = \mathbb{1}_{N \neq 0} = \begin{cases} 1 & \tilde{y}_i \neq y_i \\ 0 & otherwise \end{cases}.$$

For a given $\boldsymbol{r}, \boldsymbol{y}$, since $Z$ is determined by $\Delta$ and $B$ is determined by $N$, $I(Z; B) \leq I(\Delta; N) = C^{dc}_{\boldsymbol{r}, \boldsymbol{y}, \eta}(T_{\text{fit}})/n$. By Lemma A.1, for every Bernoulli random variables $B, Z$

$$|\mathbb{E}[Z] - \mathbb{E}[Z|B=1]| \leq \sqrt{\tfrac{1}{2}I(Z;B)}/\mathbb{E}[B]$$

And hence in our case (since $\mathbb{E}[B] = \eta$),

$$\mathbb{E}[Z] - \mathbb{E}[Z|B=1] \leq \sqrt{\frac{C^{dc}_{\boldsymbol{r}, \boldsymbol{y}, \eta}(T_{\text{fit}})}{2n}} \cdot \frac{1}{\eta}.$$

But $\mathbb{E}[Z]$ corresponds to the probability that $g(r) = y$ for $(r, y)$ in the train set, while $\mathbb{E}[Z|B=1]$ corresponds to this probability over the noisy samples. Hence the memorization gap is bounded by

$$\mathop{\mathbb{E}}_{(\boldsymbol{r},\boldsymbol{y}) \sim \mathcal{D}_r}\left[\sqrt{\frac{C^{dc}_{\boldsymbol{r}, \boldsymbol{y}, \eta}(T_{\text{fit}})}{2n}} \cdot \frac{1}{\eta}\right] \leq \frac{1}{\eta}\sqrt{\mathop{\mathbb{E}}_{(\boldsymbol{r},\boldsymbol{y}) \sim \mathcal{D}_r}\left[\frac{C^{dc}_{\boldsymbol{r}, \boldsymbol{y}, \eta}(T_{\text{fit}})}{2n}\right]} = \sqrt{\frac{C^{dc}_{\mathcal{R}, \eta}(T_{\text{fit}})}{2n}} \cdot \frac{1}{\eta}$$

using the Jensen inequality and the concavity of square root for the first inequality. $\qquad \square$

## C  POSITIVE RATIONALITY GAP LEAVES ROOM FOR IMPROVEMENT

In this appendix, we prove the "performance on the table theorem" that states that we can always transform a robust training procedure with a positive rationality gap into a training procedure with better performance:

**Theorem C.1** (*Performance on the table theorem, restated*). *For every training procedure $T$ and $\mathcal{D}_{test}, n, \eta$, if $\mathcal{D}_{train} = \mathcal{D}_{test}^n$ there exists a training procedure $S$ such that*

$$\mathsf{Test}_{S,\mathcal{D},n} \geq \mathsf{NTrain}_{T,\mathcal{D},n}(\eta) - o(1) \qquad (5)$$

*where $o(1)$ is a term that vanishes with $n$, and under the assumption that $\mathsf{Train}_{T,\mathcal{D},n}(\eta) \geq \mathsf{NTrain}_{T,\mathcal{D},n}(\eta)$.*

For any reasonable training procedure $T$, performance on noisy train samples will not be better than the overall train accuracy, and hence the assumption will be satisfied. In particular (since we can always add noise to our data), the above means that we can obtain a procedure $S'$ whose clean test performance is at least $\mathsf{Test}_T + \Delta$ where $\Delta = \mathsf{NTrain}_T(\eta) - \mathsf{Test}_T$ is the *rationality gap* of $T$. Hence if the rationality gap is larger than the robustness gap, we can use the above to improve the test performance of "irrational" networks. (Note that the robustness gap of almost all standard training procedure is at most $\eta$ and in fact often much smaller.) We stress that the procedure of Theorem 3.2, while running in "polynomial time", is not particularly practical, since it makes *inference* be as computationally expensive as training. However, it is a proof of concept that irrational networks are, to some extent, "leaving performance on the table".

*Proof.* Let $T$ be a procedure as above. Our algorithm $S$ would be the following:

- **Training:** The algorithm will not do any training but on input labels $D = \{(x_i, \tilde{y}_i)\}$ simply stores these labels.

- **Inference:** On input a data point $x$, Algorithm $S$ will choose $i \in [n]$ at random, and run $T$ on the data $D$ replacing the $i$-th sample with $(x, \tilde{y})$ where $\tilde{y}$ is chosen uniformly at random. The output is $f(x)$ where $f$ is the classifier output by $T$

First note that while the number of noisy samples could change by one by replacing $(x_i, y_i)$ with $(x, \tilde{y})$, since this number is distributed according to the Binomial distribution with mean $\eta n$ and standard deviation $\sqrt{(1-\eta)\eta n} \gg 1$, this change can affect probabilities by at most $o(1)$ additive factor. If $\mathcal{Y}$ has $k$ classes, then with probability $1 - 1/k$ we will make $(x, \tilde{y})$ noisy ($y \neq \tilde{y}$) in which case the expected performance on it will be $\mathsf{NTrain}_T(\eta)$. With probability $1/k$, we choose the correct label $y$ in which case performance on this sample will be equal to the expected performance on clean samples which by our assumptions is at least $\mathsf{NTrain}_T(\eta)$ as well. $\qquad\square$

## D    EXPERIMENTAL DETAILS

We perform an empirical study of the RRM bound for a wide variety of self-supervised training methods on the ImageNet (Deng et al., 2009) and CIFAR-10 (Krizhevsky et al., 2009) training datasets. We provide a brief description of all the self-supervised training methods that appear in our results below. For each method, we use the official pre-trained models on ImageNet wherever available. Since very few methods provide pre-trained models for CIFAR-10, we train models from scratch. The architectures and other training hyper-parameters are summarized in Table H.4 and Table H.3. Since our primary aim is to study the RRM bound, we do not optimize for reaching the state-of-the-art performance in our re-implementations. For the second phase of training, we use L2-regularized linear regression, or small non-interpolating Multi-layer perceptrons (MLPs).

### D.1    SELF-SUPERVISED TRAINING METHODS ($T_{\mathrm{PRE}}$)

There are a variety of self-supervised training methods for learning representations without explicit labels. The two chief classes of self-supervised learning methods are:

1. *Contrastive learning:* These methods seek to find an embedding of the dataset that pushes a *positive* pair of images close together and a pair of *negative* images far from each other. For example, two different augmented versions of the same image may be considered a positive pair, while two different images may be considered a negative pair. Different methods such as Instance Discrimination, MoCo, SimCLR, AMDIM, differ in the the way they select the positive/negative pairs, as well other details like the use of a memory bank or the encoder architecture. (See Falcon & Cho (2020) for detailed comparison of these methods).

2. *Handcrafted pretext tasks:* These methods learn a representation by designing a fairly general supervised task, and utilizing the penultimate or other intermediate layers of this network as the representation. Pretext tasks include a variety of methods such as predicting the rotation angle of an input image (Gidaris et al., 2018), solving jigsaw puzzles (Noroozi & Favaro, 2016), colorization (Zhang et al., 2016), denoising images (Vincent et al., 2008) or image inpainting (Pathak et al., 2016).

Additionally, adversarial image generation can be used for by augmenting a the image generator with an encoder (Donahue & Simonyan, 2019). We focus primarily on contrastive learning methods since they achieve state-of-the-art performance. We now describe these methods briefly.

**Instance Discrimination:** (Wu et al., 2018) In essence, Instance Discrimination performs supervised learning with *each* training sample as a separate class. They minimize the non-parametric softmax loss given below for the training dataset.

$$J(\theta) = -\sum_{i=1}^{n} log\left(\frac{\exp(v_i^T v/\tau)}{\sum_{j=1}^{n} \exp(v_i^T v/\tau)}\right) \tag{6}$$

where $v_i = f_\theta(x_i)$ is the feature vector for the $i$-th example. They use memory banks and a contrastive loss (also known as Noise Contrastive Estimation or NCE (Gutmann & Hyvärinen, 2010)) for computing this loss efficiently for large datasets. So in this case, a positive pair is an image and itself, while a negative pair is two different training images.

**Momentum Contrastive (MoCo):** (He et al., 2020) MoCo replaces the memory bank in Instance Discrimination with a momentum-based query encoder. MoCoV2 (Chen et al., 2020c) uses various modifications from SimCLR, like a projection head, and combines it with the MoCo framework for improved performance.

**AMDIM:** (Bachman et al., 2019) AMDIM uses two augmented versions of the same image. For these augmentations, they use random resized crops, random jitters in color space, random horizontal flip and random conversion to grayscale. They apply the NCE loss across multiple scales, by using features from multiple layers. They use a modified ResNet by changing the receptive fields to decrease overlap between positive pairs.

**CMC:** (Tian et al., 2019) CMC creates two views for contrastive learning by converting each image into the Lab color space. L and ab channels from the same image are considered to be a positive pair, while those from two different images are considered to be a negative pair.

**PiRL:** (Misra & Maaten, 2020) PiRL first creates a jigsaw transformation of an image (it divides an image into 9 patches and shuffles these patches). It treats an image and its jigsaw as a positive pair, and that of a different image as a negative pair. They additionally modify encoder on the jigsaw branch.

**SimCLRv1 and SimCLRv2:** (Chen et al., 2020a;b) SimCLR also use strong augmentations to create positive and negative pairs. They use random resized crops, random Gaussian blur and random jitters in color space. Crucially, they use a projection head that maps the representations to a 128-dimensional space where they apply the contrastive loss. They do not use a memory bank, but use a large batch size.

**InfoMin:** InfoMin uses random resized crop, color jitter and gaussian blur, as well as jigsaw shuffling from PiRL.

### D.2    SIMPLE CLASSIFIER ($T_{\text{FIT}}$)

After training the representation learning method, we extract representations $r$ for the training and test images. We do not add random augmentations to the training images (unless stated otherwise). Then, we train a simple classifier on the dataset $\{r(x_i), y_i\}_{i=1}^n$. We use a linear classifier in most cases, but we also try a small multi-layer perceptron (as long as it has few parameters and does not interpolate the training data). We add weight decay in some methods to achieve good test accuracy (See Table H.4 for values for each method.) For the noisy experiment, we set the noise level to $\eta = 5\%$. To compute the complexity bound $C^{dc}$ we run 20 trials of the noisy experiment for CIFAR-10 and 50 trials for ImageNet.

### D.3    EXPERIMENTAL DETAILS FOR EACH PLOT

**Figure 1.** This figure shows the robustness, rationality and memorization gap for various SSS algorithms trained on CIFAR-10. The type of self-supervised method, the encoder architecture, as well

as the training hyperparameters are described in Table H.3. For the second phase $T_{\text{fit}}$, we use L2-regularized linear regression for all the methods. For each algorithm listed in Table H.3, the figure contains 2 points, one without augmentations, and one with augmentations. Further, we compute the complexity measure $\mathsf{C}^{\text{dc}}$ for all the methods. All the values (along with the test accuracy) are listed in Table H.1.

**Figure 2.** This figure shows the robustness, rationality and memorization for CIFAR-10 for all the same methods as in Figure 1. We only include the points without augmentation to show how rationality behaves when $(\mathcal{D}_{\text{train}}, \mathcal{D}_{\text{test}})$ are identical. All the values (along with the test accuracy) are listed in Table H.1. For the supervised architectures, we train a Myrtle-5 (Page, 2018) convolutional network, a ResNet-18 (He et al., 2016) and a WideResNet-28-10 (Zagoruyko & Komodakis, 2016) with standard training hyperparameters.

**Figure 3 and Figure 4.** These figures show the robustness, rationality and memorization for the ImageNet dataset. The type of self-supervised method, the encoder architecture, as well as the training hyperparameters are described in Table H.4. For the second phase $T_{\text{fit}}$, we use L2-regularized linear regression for all the methods. The figures also contain some points with 10 augmentations per training image. Further, we compute the complexity measure $\mathsf{C}^{\text{dc}}$ for all three methods - SimCLRv2 with architectures ResNet-50-1x and ResNet-101-2x. All the values (along with the test accuracy) are listed in Table H.2.

**Figure 5** This figure shows the effect of increasing augmentations. We add $t = \{2, ..., 10\}$ augmentations and re-train the simple classifier. We do this for the CIFAR-10 dataset, AMDIM self-supervised training with the AMDIM encoder and linear regression (See Table H.3 for the hyperparameters).

### D.4 ADDITIONAL RESULTS

#### D.4.1 GENERALIZATION ERROR OF SSS ALGORITHMS

To show that SSS algorithms have qualitatively different generalization behavior compared to standard end-to-end supervised methods, we repeat the experiment from Zhang et al. (2017). We randomize all the training labels in the CIFAR-10 dataset and train 3 high-performing SSS methods on these noisy labels. For results see Table D.1. Unlike fully supervised methods, SSS algorithms do not achieve 100% training accuracy on the dataset with noisy labels. In fact, their training accuracies are fairly low ($\approx$ 15-25%). This suggests that the empirical Rademacher complexity is bounded. The algorithms were trained without any augmentations during the simple fitting phase for both SSS and supervised algorithms. The SSS methods were trained using parameters described in Table H.3.

**Table D.1** – Empirical Rademacher for fully supervised vs. SSS algorithms on CIFAR-10, Averaged over 5 runs. No augmentations were added.

| Training method | Architecture/Method | Train Acc | Test Acc |
|---|---|---|---|
| Supervised (Zhang et al., 2017) | Inception (no aug) | 100% | 86% |
| | (fitting random labels) | **100%** | 10% |
| SSS | SimCLR (ResNet-50) + Linear | 94% | 92% |
| | (fitting random labels) | **22%** | 10% |
| | AMDIM (AMDIM Encoder) + Linear | 94% | 87.4% |
| | (fitting random labels) | **18%** | 10% |
| | MoCoV2 (ResNet-18) + Linear | 69% | 67.6% |
| | (fitting random labels) | **15%** | 10% |

### D.5 RRM BOUND WITH VARYING NOISE PARAMETER

We now investigate the effect of varying noise levels on the three gaps as well as on the complexity. We see that the robustness gap increases as we add more noise—this is expected as noise should affect the clean training accuracy. We also observe that the memorization gap decreases, suggesting that $\mathsf{C}^{\text{dc}}_{\eta}$ as a function of $\eta$ goes down faster than $\eta^2$ (see Appendix B). The Theorem II bound on memorization gap also decays strongly with the $\eta$, becoming more tight as the noise increases.

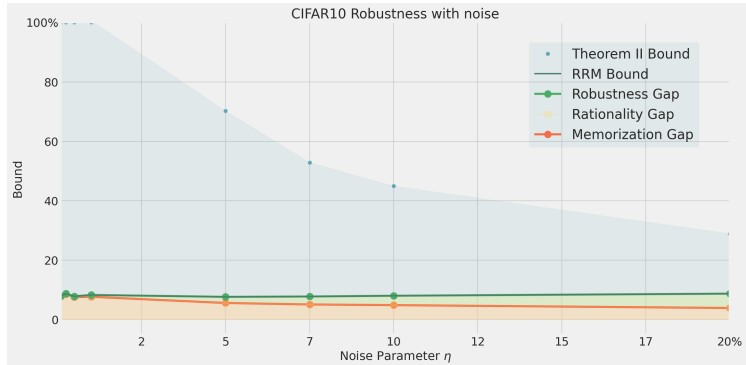

**Figure D.1** – RRM + bound with changing $\eta$

### D.5.1 CONVERGENCE OF COMPLEXITY MEASURES

We now plot the complexity measures $C^{dc}$ and $C^{pc}$ with increasing number of trials for one of the SSS algorithms. As expected, $C^{dc} < C^{pc}$ and $C^{dc}$ converges in about 20 trials for CIFAR-10. On the other hand, the complexity computations for ImageNet need many more trials for convergence, since it contains about 10 augmentations $\times 1.2$ million training samples making it cost prohibitive to compute for all the methods. For the CIFAR-10, we use AMDIM with the AMDIM encoder architecture without augmentations. For ImageNet, we use SimCLRv2 with the ResNet-101 architecture with 10 augmentations per training sample.

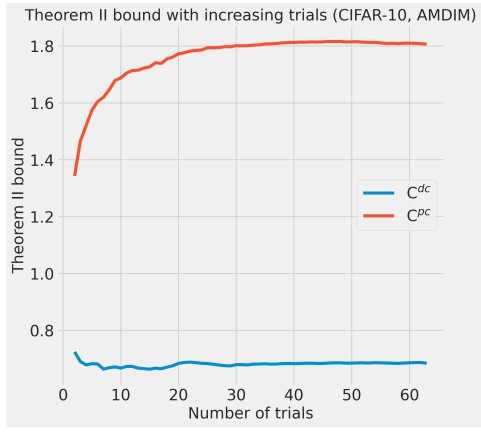

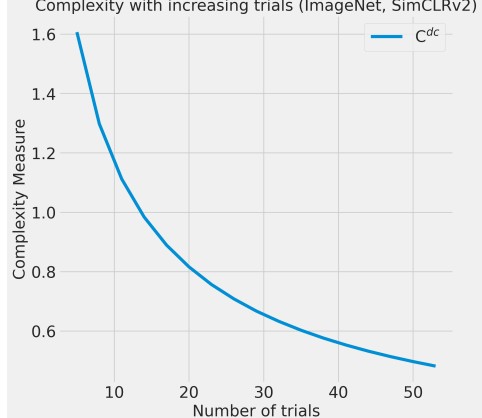

**(a)** Theorem II bound with increasing trials. The bound based on $C^{dc}$ is lower than $C^{pc}$ as expected, and converges within 20 trials.

**(b)** Theorem II bound with increasing trials. $C^{dc}$ is slow to converge due to the large dataset size (10 augmentations $\times$ 1.2 million training samples).

**Figure D.2** – Convergence of Theorem II bounds for CIFAR-10 and ImageNet

## E EXAMPLES OF ALGORITHMS WITH LARGE GAPS

While we argued that SSS algorithms will tend to have small robustness, rationality, and memorization gaps, this does not hold in the worst case and there are examples of such algorithms that exhibit large gaps in each of those cases.

### E.1 LARGE ROBUSTNESS GAP

Large robustness gap can only arise via computational (as opposed to statistical) considerations. That is, if a training procedure outputs a classifier $f \in \mathcal{F}$ that achieves on average accuracy $\alpha$ on a

clean train set $(X, Y)$, then with high probability, if $(X, \tilde{Y})$ is an $\eta$-noisy train set then *there exists* $f \in \mathcal{F}$ that achieves $\alpha(1 - \eta)$ accuracy on this train set (by fitting only the "clean" points).

However, the training algorithm might not always be able to find such a classifier. For example, if the distribution has the form $(x, y) = (x, \sum a_j x_j \mod 2)$ where $x \sim GF(2)^{\ell} = \mathbb{Z}_2^{\ell}$ and $a \in GF(2)^{\ell}$ is some hidden vector, then there is an efficient algorithm (namely Gaussian elimination) to find $a$ given the samples $(x, y)$ and hence get accuracy 1. However, for every $\varepsilon > 0$ and $\eta > 0$, there is no known efficient algorithm that, given a $1 - \eta$ perturbed equations of the form $\{\langle a, x_i \rangle = \tilde{y}_i\}_{i \in [n]}$ finds $a' \in GF(2)^{\ell}$ such that $\sum a'_j x_j = \sum a_j x_j \mod 2$ on a $1/2 + \varepsilon$ fraction of the $x$'s. This is known as the *learning parity with noise (LPN)* problem (Blum et al., 1993).

The assumption of robustness is *necessary* for a small generalization gap, in the sense that we can come up with (contrived) examples of algorithms that have small rationality and memorization gaps while still having large generalization gap. For example, consider an algorithm $T$ that has large generalization gap (high train accuracy and small test accuracy) , and suppose we augment to the following algorithm

$$T'(\boldsymbol{x}, \boldsymbol{y}) = \begin{cases} T(\boldsymbol{x}, \boldsymbol{y}) & \text{if } \boldsymbol{y} \text{ is "clean"} \\ 0 & \text{if } \boldsymbol{y} \text{ is "noisy"} \end{cases}$$

where $0$ denotes the constant zero function (e.g., some trivial classifier) and we use some algorithm to estimate whether or not the labels are noisy. (Such estimates can often be achieved in many natural cases.) The algorithm $T'$ will inherit the generalization gap of $T$, since that depends only on the experiment without noise. Since performance on noisy and clean training samples will be the same (close to random), will have zero memorization gap. Since we have assumed small test accuracy, it will have zero rationality gap also.

### E.2 LARGE RATIONALITY GAP

As discussed in Section C, in the case that $\mathcal{D}_{\text{train}} = \mathcal{D}_{\text{test}}^n$, a robust algorithm with large rationality gap leaves "performance on the table". We can obtain such algorithms by artificially dropping performance on the test data. For example, in the SSS framework, since the representation $r$ is over-parameterized and can memorize the entire train set, we can consider the trivial representation

$$r(x) = \begin{cases} x & x \text{ in train set} \\ 0 & \text{otherwise} \end{cases}$$

If we now train some simple classifier on $r(x)$ then it can have non-trivial performance on the noisy train samples, while getting trivial accuracy on all samples outside the train set.

In cases where $\mathcal{D}_{\text{train}}$ and $\mathcal{D}_{\text{test}}$ are different (for example when $\mathcal{D}_{\text{train}}$ is an augmented version of $\mathcal{D}_{\text{test}}$) then we can no longer claim that a large rationality gap corresponds to "leaving performance on the table". For example, we do observe (mild) growth in the rationality gap as we add more augmented points to the training set.

### E.3 LARGE MEMORIZATION GAP

It is not hard to find examples of networks with large memorization gap. Indeed, as mentioned before, any standard interpolating supervised learning algorithm will get a memorization gap close to 1.

## F ROBUSTNESS OF LEAST SQUARES CLASSIFIERS

One can prove robustness for classes of algorithms under varying assumptions. As a simple example, we record here a self-contained observation of how margin leads to robustness in least squares minimization. (We believe that this bound is folklore, but weren't able to find the right reference.) This is a very simple but also pessimistic bound, and much better ones often hold.

**Lemma F.1 .** *Let $x_1, \ldots, x_n \in \mathbb{R}^d$ and $y_1, \ldots, y_n \in [k]$, and consider a linear function $f : \mathbb{R}^d \to \mathbb{R}^k$ that minimizes the quantity $\sum_{i \in [n], j \in [k]} |f(x_i)_j - \mathbb{1}_{y_i=j}|^2$, and suppose that for p fraction of the i's, the maximum over $j \in [k]$ of $f(x_i)$ is $\gamma$ larger than the second-largest value.*

*Then in expectation, if we let $\tilde{y}$ be the $\eta$-noisy version of $y$ and $\tilde{f}$ minimizes $\sum_{i \in [n], j \in [k]} |\tilde{f}(x_i)_j - \mathbb{1}_{\tilde{y}_i=j}|^2$, we get that $\arg \max_j \tilde{f}(x_i) = y_i$ for at least $p - 4\eta/\gamma^2$ fraction of the i's.*

*Proof.* We identify $y$ with its "one hot" encoding as a vector in $\mathbb{R}^{nk}$. Let $V \subseteq \mathbb{R}^{nk}$ be the subspace of all vectors of the form $(g(x_1), \ldots, g(x_n))$ for linear $g : \mathbb{R}^d \to \mathbb{R}^k$. If $f$ is the minimizer in the theorem statement, and $p = (f(x_1), \ldots, f(x_n))$ then $p = \Pi_V y$ where $\Pi_V$ is the orthogonal projection to the subspace $v$. If $\tilde{f}$ is the minimizer for the noisy labels and $\tilde{p} = (\tilde{f}(x_1), \ldots, \tilde{f}(x_n))$, then $\tilde{p} = \Pi_V \tilde{y} = \Pi_V(y + e)$ where $e$ is the noise vector $\tilde{y} - y$.

Hence $\|p - \tilde{p}\| = \|\Pi_V e\| \leq \|e\|$. But in expectation $\|e\|^2 \leq 2\eta n$ (since we flip a label with probability $\leq \eta$). For every point $i$ for which the margin was at least $\gamma$ in $p$, if $\tilde{p}$'s prediction is different in $i$, then the contribution of the $i$-th block to their square norm difference is at least $\gamma^2/2$ (by shifting the maximum coordinate by $-\gamma/2$ and the second largest one by $\gamma/2$). Hence at most $4\eta n/\gamma^2$ of these points could have different predictions in $p$ and $\tilde{p}$ $\qquad\square$

## G ROBUSTNESS OF EMPIRICAL RISK MINIMIZER

The (potentially inefficient) algorithm that minimizes the classification errors is always robust.

**Lemma G.1 .** *Let $T(x, y) = \arg \min_{f \in \mathcal{F}} \sum_{i=1}^n \mathbb{1}_{f(x_i) \neq y_i}$. Then for every $\eta > 0$,*

$$\text{Robustness gap}(T) \leq 2\eta \ .$$

*Proof.* Let $x, y$ be any train set, and let $\alpha = \min_{g \in \mathcal{F}} \sum_{i=1}^n \mathbb{1}_{g(x_i) \neq y_i}$ and $f$ be the minimizer of this quantity. Let $\tilde{y}$ be the $\eta$-noisy version of $y$ and let $\tilde{\eta}$ be the fraction of $i$ on which $y_i \neq \tilde{y}_i$. Then,

$$\sum_{i=1}^n \mathbb{1}_{f(x_i) \neq y_i} \leq \alpha + \tilde{\eta} \ . \tag{7}$$

Hence if $\tilde{f}$ is the minimizer of (7) then we know that $\tilde{f}(x_i) \neq \tilde{y}_i$ for at most $\alpha + \tilde{\eta}$ fraction of the $i$'s, and so $\tilde{f}(x_i) \neq y_i$ for at most $\alpha + 2\tilde{\eta}$ fraction of the $i$'s. Since the train accuracy of $T$ is $1 - \alpha$ and in expectation of $\tilde{\eta}$ is $\eta$, we get that in expectation

$$\text{Train}_T(\eta) \geq \text{Train}_T - 2\eta$$

$\square$

# H    LARGE TABLES

**Table H.1** – Summary of all the methods, architectures and the corresponding results (gaps and accuracies) on CIFAR-10, sorted by generalization gap. While Figure 1 already plots this data, here we also provide the test performance of the corresponding models.

| Method | Backbone | Data Aug | Generalization Gap | Robustness | Memorization | Rationality | Theorem II bound | RRM bound | Test Acc |
|--------|----------|----------|-------------------|------------|--------------|-------------|------------------|-----------|----------|
| mocov2 | resnet18 | True | -7.35 | 0.07 | 0.21 | 0.00 | 3.47 | 0.28 | 67.19 |
| mocov2 | wide_resnet50_2 | True | -6.37 | 0.18 | 1.03 | 0.00 | 7.63 | 1.21 | 70.99 |
| mocov2 | resnet101 | True | -6.01 | 0.15 | 0.71 | 0.00 | 6.38 | 0.86 | 68.58 |
| mocov2 | resnet50 | True | -5.38 | 0.19 | 0.84 | 0.00 | 6.99 | 1.03 | 69.68 |
| simclr | resnet50 | True | -2.89 | 0.30 | 0.55 | 0.00 | 6.63 | 0.85 | 91.96 |
| amdim | resnet101 | True | -0.91 | 0.64 | 3.70 | 0.00 | 25.99 | 4.34 | 63.56 |
| amdim | resnet18 | True | 0.33 | 0.23 | 1.15 | 0.00 | 8.66 | 1.38 | 62.84 |
| mocov2 | resnet18 | False | 1.43 | 0.15 | 1.24 | 0.03 | 14.14 | 1.43 | 67.60 |
| simclr | resnet18 | False | 1.43 | 0.28 | 0.79 | 0.36 | 13.35 | 1.43 | 82.50 |
| amdim | wide_resnet50_2 | True | 1.60 | 0.69 | 2.46 | 0.00 | 19.20 | 3.15 | 64.38 |
| simclr | resnet50 | False | 1.97 | 0.22 | 0.78 | 0.97 | 15.75 | 1.97 | 92.00 |
| simclr | resnet50 | False | 2.24 | 0.52 | 1.71 | 0.01 | 19.53 | 2.24 | 84.94 |
| mocov2 | resnet50 | False | 2.72 | 0.30 | 2.96 | 0.00 | 24.18 | 3.26 | 70.09 |
| mocov2 | resnet101 | False | 2.82 | 0.33 | 3.03 | 0.00 | 22.78 | 3.36 | 69.08 |
| mocov2 | wide_resnet50_2 | False | 3.11 | 0.38 | 2.79 | 0.00 | 22.39 | 3.18 | 70.84 |
| amdim | resnet50_bn | True | 3.69 | 0.84 | 4.22 | 0.00 | 31.12 | 5.06 | 66.44 |
| amdim | resnet18 | False | 4.34 | 0.42 | 4.58 | 0.00 | 33.47 | 5.00 | 62.28 |
| amdim | amdim_encoder | True | 4.43 | 0.68 | 0.36 | 3.39 | 10.32 | 4.43 | 87.33 |
| amdim | amdim_encoder | False | 6.68 | 2.08 | 5.69 | 0.00 | 70.52 | 7.77 | 87.38 |
| amdim | resnet101 | False | 12.46 | 1.22 | 14.26 | 0.00 | 100.00 | 15.49 | 62.43 |
| amdim | wide_resnet50_2 | False | 13.07 | 1.70 | 15.33 | 0.00 | 100.00 | 17.03 | 63.80 |
| amdim | resnet50_bn | False | 14.73 | 1.81 | 16.63 | 0.00 | 100.00 | 18.43 | 66.28 |

**Table H.2** – Summary of all the methods, architectures their corresponding results (gaps and accuracies) on ImageNet, sorted by generalization gap. While Figure 4 already plots this data, here we also provide the test performance of the corresponding models.

| Method | Backbone | Data Aug | Generalization Gap | Robustness | Mem- orization | Rationality | Theorem II bound | RRM bound | Test Acc |
|---|---|---|---|---|---|---|---|---|---|
| simclrv2 | r50_1x_sk0 | True | -2.34 | 0.26 | 0.68 | 0.00 | 46.93 | 0.94 | 70.96 |
| simclrv2 | r101_2x_sk0 | True | 0.63 | 0.10 | 0.80 | 0.00 | 47.90 | 0.91 | 77.24 |
| simclrv2 | r152_2x_sk0 | True | 1.00 | 0.13 | 0.77 | 0.10 | NA | 1.00 | 77.65 |
| moco | ResNet-50 | True | 1.32 | 0.57 | 0.93 | 0.00 | NA | 1.49 | 70.15 |
| InfoMin | ResNet-50 | True | 4.88 | 0.81 | 1.01 | 3.06 | NA | 4.88 | 72.29 |
| PiRL | ResNet-50 | True | 6.23 | 0.29 | 0.99 | 4.95 | NA | 6.23 | 60.56 |
| InsDis | ResNet-50 | True | 6.85 | 0.25 | 1.13 | 5.46 | NA | 6.85 | 58.30 |
| simclrv2 | r101_1x_sk1 | False | 8.23 | 0.71 | 4.66 | 2.86 | NA | 8.23 | 76.07 |
| InfoMin | ResNet-50 | False | 10.21 | 2.34 | 8.96 | 0.00 | NA | 11.31 | 70.31 |
| simclrv2 | r152_1x_sk0 | False | 10.32 | 1.12 | 6.93 | 2.26 | NA | 10.32 | 74.17 |
| simclrv2 | r101_1x_sk0 | False | 10.53 | 1.11 | 6.99 | 2.42 | NA | 10.53 | 73.04 |
| simclrv2 | r50_1x_sk0 | False | 10.62 | 0.99 | 7.31 | 2.31 | NA | 10.62 | 70.69 |
| moco | ResNet-50 | False | 10.72 | 1.82 | 7.86 | 1.04 | NA | 10.72 | 68.39 |
| simclrv2 | r152_2x_sk0 | False | 10.92 | 0.75 | 7.45 | 2.72 | NA | 10.92 | 77.25 |
| simclrv2 | r101_2x_sk0 | False | 11.02 | 0.74 | 7.51 | 2.78 | NA | 11.02 | 76.72 |
| simclr | ResNet50_1x | False | 11.07 | 1.22 | 7.73 | 2.13 | NA | 11.07 | 68.73 |
| simclrv2 | ResNet-50 | False | 11.16 | 0.64 | 7.67 | 2.85 | NA | 11.16 | 74.99 |
| PiRL | ResNet-50 | False | 11.43 | 1.49 | 8.26 | 1.68 | NA | 11.43 | 59.11 |
| InsDis | ResNet-50 | False | 12.02 | 1.40 | 8.52 | 2.10 | NA | 12.02 | 56.67 |
| amdim | ResNet-50 | False | 13.62 | 0.90 | 9.72 | 3.01 | NA | 13.62 | 67.69 |
| CMC | ResNet-50 | False | 14.73 | 2.30 | 12.30 | 0.13 | NA | 14.73 | 54.60 |
| bigbigan | ResNet-50 | False | 29.60 | 3.13 | 25.19 | 1.27 | NA | 29.60 | 50.24 |

**Table H.3** – Summary of training methods with their hyper-parameters for CIFAR-10

| Self- supervised method | Backbone Architectures | Self-supervised Training | Evaluation | Simple Phase Optimization |
|---|---|---|---|---|
| AMDIM | AMDIM Encoder ResNet-18 ResNet-50 WideResNet-50 ResNet 101 | PLB Default parameters | Linear | Adam $\beta_1 = 0.8\ \beta_2 = 0.999$ Constant LR = 2e-4 Batchsize = 500 Weight decay = 1e-6 |
| MoCoV2 | ResNet-18 ResNet-50 WideResNet-50 ResNet 101 | PLB Default parameters | Linear | Adam $\beta_1 = 0.8\ \beta_2 = 0.999$ Constant LR = 2e-4 Batchsize = 500 Weight decay = 1e-6 |
| SimCLR | ResNet-18 ResNet-50 | Batchsize = 128 Epochs 200 | Linear | SGD Momentum = 0.9 Constant LR = 0.1 Weight decay 1e-6 |
| | ResNet-50 | Batchsize = 512 Epochs 600 | | |

**Table H.4** – Summary of training methods with their hyper-parameters for ImageNet

| Self-supervised method | Backbone Architecture | Pre-trained Model | Evaluation | Optimization | Weight Decay | Epochs |
|---|---|---|---|---|---|---|
| Instance Discrimination | ResNet-50 | PyContrast | Linear | SGD
Momentum = 0.9
Initial LR = 30
LR drop at {30}
by factor 0.2 | 0 | 40 |
| MoCo | ResNet-50 | Official | Linear | SGD
Momentum = 0.9
Initial LR = 30
LR drop at {30}
by factor 0.2 | 0 | 40 |
| PiRL | ResNet-50 | PyContrast | Linear | SGD
Momentum = 0.9
Initial LR = 30
LR drop at {30}
by factor 0.2 | 0 | 40 |
| CMC | ResNet-50 | PyContrast | Linear | SGD
Momentum = 0.9
Initial LR = 30
LR drop at {30}
by factor 0.2 | 0 | 40 |
| AMDIM | AMDIM Encoder | Official | Linear | SGD
Momentum = 0.9
Initial LR = 30
LR drop at {15, 25}
by factor 0.2 | 1e-3 | 40 |
| BigBiGAN | ResNet-50 | Official | Linear | SGD
Momentum = 0.9
Initial LR = 30
LR drop at {15, 25}
by factor 0.2 | 1e-5 | 40 |
| SimCLRv1 | ResNet-50 1x
ResNet-50 4x | Official | Linear | SGD
Momentum = 0.9
Constant LR = 0.1 | 1e-6 | 40 |
| SimCLRv2 | ResNet-50 1x SK0
ResNet-101 2x SK0
ResNet-152 2x SK0
ResNet-152 3x SK0 | Official | Linear | SGD
Momentum = 0.9
Constant LR = 0.1 | 1e-6 | 40 |

