# OpenReview forum: "For self-supervised learning, Rationality implies generalization, provably"
_ICLR.cc/2021/Conference — ICLR 2021 Poster_

### Official Review · AnonReviewer1 · 2020-10-13
**Not rigorous with poor presentation.**

**Rating:** 3
**Confidence:** 4

**Review:**


The authors propose to upper bound the generalization gap via three quantities, namely robustness gap, rationality gap and memorization gap, shows that the memorization gap can be bounded via standard learning theory arguments, and empirically show that all of the three terms are small. The authors also argue that if the rationality gap is large, then the performance can be improved.

1. First of all, I think this paper is highly over-claimed. I don’t see how the proposed methods provably indicates the generalization. In fact, there is no theoretical conclusion on bounding the rationality gap and little theoretical discussion on robustness gap. Instead, the authors only show the empirical estimation on the robustness gap and  rationality gap. I would like to say, such inaccurate claim makes me feel uncomfortable.

2. I would like to argue that, we cannot know the exact rationality gap, as we don’t have the data distribution at any time, thus we need a generalization bound to describe the performance of algorithm on unseen data. How do the authors deal with the rationality gap? I don’t feel empirical estimation on ‘test set’ is an acceptable choice, as ‘test set’ is only a batch of sample of real data distribution. The bound proposed by authors is not a generalization bound, thus it is meaningless to talk about the bound is vacuous or not.

3. Moreover, as we don’t know the exact rationality gap, the claim by Theorem 3.1 is also not meaningful. In other words, even if we find the rationality gap is large when evaluating on test data, what we really do is tuning the model using the test data, not improving the performance of the model on data distribution.

4. The authors argue in the abstract that the bound is independent of the complexity of the representation. However, several properties of the representation, e.g. the dimension, will definitely influence the generalization bound. I don’t feel this argument well-supported.

Overall, the decomposition itself may motivate new idea on improving the current algorithms. However, theoretically, I don’t think this paper is a rigorous paper considering the generalization bound. If the authors want to argue the decomposition have some insight on improving algorithm, the authors should focus more on the intuition, algorithm design and empirical justification. If the authors want to argue the decomposition indicate tight generalization bound, then the authors should give rigorous proof on the bound of all three terms and calculate the bound based on the theoretical prediction instead of empirical simulation. There can be some misunderstanding on some of the points in the paper, but overall, with the current presentation, I think this paper is not ready for acceptance.

---

> ### Author Response · Authors · 2020-11-11
> **Author response**
>
> Thank you for your review. We believe that some of the points raised are addressed in our response above and we will definitely make sure to clarify these better in our revisions. In particular, the main theoretical contribution of this paper is, as stated in the abstract, a rigorous bound on the generalization gap of SSS algorithms modulo certain assumptions. In particular our assumptions are that the rationality gap and the robustness gap are bounded by some small constants. These assumptions are incomparable with those used in other generalization-bounds literature, but they are both non-trivial (in the sense that they don’t, by themselves, imply a small generalization bound without the complexity bound on the classifier) and are justified by our empirical studies.
>
> - “we cannot know the exact rationality gap”: We do not need to know the exact rationality gap for our theorem to hold, we only need to know that it is bounded by a small constant. The rationality gap can be estimated using a held-out set, even when we do not have access to the entire distribution. The estimate will get better as we use more held-out samples.
> - “claim by Theorem 3.1 is not meaningful”: We argue that claim 3.1 is meaningful since it does not require the label of the test sample. As described in the proof for the theorem (Appendix C), when the algorithm is asked to make a prediction for any new test sample x, we can include this sample in the train set with a random label, and this would increase the probability of getting the correct label for this test sample if the rationality gap is high. This process can be applied for every new sample from the data distribution even when we do not have access to its label.
> - “dependence on properties of representation”: Our bound does indeed, as stated in the abstract, depend only on the complexity of the simple classifier. However, you are right that the dimension of the representation can indirectly impact the bound, since (for example) a linear classifier with larger input will be more complex than a linear classifier with smaller input. We will add this comment to this paper - thank you!

---

> > ### Comment · AnonReviewer1 · 2020-11-16
> > **Thanks for your response. I would like to re-emphasize some of my ideas.**
> >
> > I guess there are some different understanding on the theoretical analysis. So I would like to re-emphasize some of my ideas in the review.
> >
> > 1. I still don't think the "provably" argument is not over-claimed. From the authors' response I guess the authors think that we can assume rationality gap and robustness gap is bounded, which we definitely not. Our assumptions should be $\textbf{Falsifiable}$. In practice we need to have the ability to check if one assumption holds or not. However, rationality gap cannot be tested, as I said, we don't have the testing distribution, we can never know this rationality gap. What we can do, is theoretically bounded the gap, which is the main focus of the learning theory. I would rather not treating this kinds of conclusion as the generalization bound. In fact, if the authors give some generalization bound that does not based on the assumption that this rationality gap is small, or this is an empirical paper that shows the rationality gap is closely related to the generalization, I would be happy to accept that.
> > 2. From the above point, as the rationality gap is not available in practice, we cannot know when to use the Theorem 3.1. Thus I say Theorem 3.1 is not meaningful. More or less this is like the Schrodinger's cat, if we cannot observe it, then we don't know what happens. We should have an indicator to tell us when can we use Theorem 3.1. And use the test data is not a choice for the indicator.
> >
> > Hope this clarification helps the authors understand my ideas and I'm happy to have more discussion on the availability of rationality gap.

---

> > > ### Author Response · Authors · 2020-11-16
> > > **Thanks for the response. We believe the main issue is a difference of philosophies rather than us over-claiming.**
> > >
> > > It seems that some of the issues are "philosophical“. We think there is room for more than one type of a theory paper. In contrast, it seems that your definition of a “provable generalization bound” is that it is a function of some quantity such as the norms of weights that we can measure based only on the classifier and training set. These types of generalization bounds have generally not been very successful for deep learning. We believe our notion of rationality is a good way forward.
> > >
> > > It is **always** possible to remove this assumption by using our “performance on the table” theorem. This transformation can always be used (and so it’s not like “Schrödinger’s cat”) and will ensure that the generalization gap of the new classifier is just the robustness and memorization gaps of the old one. However, we do not recommend this for practical purposes because it makes inference as computationally expensive as training. That said, in practice the assumption that rationality is not more than a few percent is very justifiable (and of course can always be tested against a validation set).
> > >
> > > Last, regardless of the “philosophical” views (of which each reader can make their own mind) we do think we are very clear in the paper about what we prove, what we demonstrate empirically, and what we assume. For instance, even in title of the paper we say **rationality implies generalization** which could only be interpreted as: if you are willing to **assume** rationality you can prove generalization (i.e., small generalization gap).

---

### Official Review · AnonReviewer3 · 2020-10-28
**The paper proposes a 3-term decomposition of the generalization gap**

**Rating:** 4
**Confidence:** 4

**Review:**

The paper analyzes the generalization gap for self-supervised learning. This paper's contribution includes the proposal of decomposing the generalization bound into three terms: robustness, rationality, and memorization (RRM). The three terms explain the generalization gap with some different perspectives. With the RRM decomposition's help, it proves that since SSS doesn't memorize data, small robustness and small rationality gap will naturally guarantee a small generalization gap.

Although I believe the RRM is novel and might bring some good insights for future studies, in isolation to self-supervised learning, this paper's main results (on the SSS part) seem to be stating something obvious in a fancy way. It is well understood that self-supervised learning has a small generalization gap, considering downstream tasks only learn from a function class with small capacity. For each T_pre, the generalization gap is guaranteed to be small with uniform convergence. The remark 3.2 doesn't really make sense to me. As for SSS, the T_pre is obtained with x sampled from the marginal distribution (with much more dataset), and T_fit is trained from (x,y) pairs generated from the joint distribution. T_pre is not supposed to see the same training examples, especially y. Therefore T_pre should not be memorizing the data samples.

The prior work mentioned in this paper in understanding SSL targets to explain why the representation learned from pretext tasks can be useful for the downstream task, hence proving a small generalization error (instead of generalization gap), which is more important in understanding the success of self-supervised learning. And assumptions like (approximate) conditional independence were only to show a small approximation error and were not needed for proving a small generalization gap. Therefore removing these assumptions does not seem to be a real contribution here.

It will be more interesting to me if the paper focuses more on the RRM decomposition and gives some further insights into how to use these terms in the future to improve the generalization performance of certain algorithms or tasks.

---

> ### Author Response · Authors · 2020-11-11
> **Author response**
>
> Thank you for your review. Your review points out that we could be more clear in explaining our results and we will make sure to do so in the next revision. As we discussed above, it is not well known that self-supervised learning has a small generalization gap. Small generalization gap can be expected when the downstream tasks use disjoint data from that used to learn the representation. However, our paper considers the “data reuse” case (as mentioned by AnonReviewer4) , where we use the same dataset for both learning representation and classifying. This is also the standard setup used by many applied works which show SSS algorithms can achieve near state-of-art performance on datasets such as ImageNet and CIFAR-10 (e.g.: The current state-of-the-art method SimCLR [1] reuses the same data between representation learning and linear evaluation). For this data reuse case, generalization bounds were not known before, and indeed are not even true without making some assumptions. The contribution of our paper is both to prove that there are assumptions that imply small generalization gap, as well as to show that these assumptions actually hold for various practical algorithms.
>
> We argue that our work also helps us understand the test error of self-supervised learning in the following sense --- as long as the algorithm achieves high training accuracy and satisfies our assumptions, it can be guaranteed to have high test accuracy for simple downstream classifiers. This allows us to decouple the goal of high training accuracy (which we can optimize empirically) and small generalization gap.
>
> References:
> [1] Chen, T., Kornblith, S., Norouzi, M. and Hinton, G., 2020. A simple framework for contrastive learning of visual representations. arXiv preprint arXiv:2002.05709.

---

### Official Review · AnonReviewer4 · 2020-10-29
**Innovative paper with clear presentation**

**Rating:** 7
**Confidence:** 4

**Review:**

The present paper aims to understand the generalization capability of self-supervised learning algorithms that fine-tune a simple linear classifier to the labels. Analyzing generalization in this case is challenging due to a data re-use problem: the same training data that is used for self-supervised learning is also used to fit the labels. The paper addresses this issue by implicitly conditioning on the training covariates x and then deriving generalization bounds that depend only on (hypothetical) noise to the labels y. The paper show that, empirically, the dominant factor in generalization error is a certain quantity called the "memorization gap", which can also be upper-bounded via theoretical analysis (the theoretical bound seems to be loose by about a factor of 4 compared to the empirical measurement, but is still non-vacuous in many cases). Interestingly, this is *not* the case for standard supervised learning, likely to the higher-complexity models used to fit the labels; in that case the memorization gap is high, but a different gap (called the "rationality gap") is large in magnitude but negative.

Overall, the paper is clearly presented, innovative, and has interesting empirical and theoretical results. It seems like a clear accept to me, with my only uncertainty that I am not completely familiar with the related literature. I am also not sure why the authors could not use the Rademacher complexity---are there theoretical obstacles to using it to upper-bound generalization error in this setting, or is the problem that it is too large? If the latter, then have you considered using your approach in settings other than just the self-supervised setting in order to improve on Rademacher complexity bounds?

====

Framing comments / questions:

-I don't like the word rationality, since it has a technical meaning in Bayesian statistics that is not the same as the usage here (i agree they are somewhat similar in flavor, but I think it's confusing to conflate them).

-I'm not sure it's correct to say that SS+S is a dominant methodology. In practice we would almost always do full fine-tuning on the self-supervised representation, rather than just the final layer. Still, starting with final layer fine-tuning is a reasonable start for analysis.

-It seems an important point of your analysis is that we can condition on x and then just look at label noise for measuring generalization. It seems like empirical Rademacher complexity bounds also condition on x, so is there a fundamental difference here? (I think you try to address this in Remark 3.3 but I didn't understand your point there.)

======

A few presentation comments:

-I didn't understand this claim: " An optimal Bayesian procedure would have zero rationality gap, and indeed this gap is typically zero or small in practice."

-Drawing lines between the dots (and shading the area under the curve) in Figure 1 is inappropriate, since the different points don't follow a logical linear progression (it's really just a scatter plot).

-In Fact I, why do we need to take the max with zero? The result is still true even without the max, I believe.

-In Fact I, it would be helpful to comment on the effect of changing eta. Do we expect certain of these quantities to get bigger or smaller in that case? Any heuristic intuition for how to choose the best eta?

-Section 2.1 is a bit dense.

-I liked Figure 2 a lot.

---

> ### Author Response · Authors · 2020-11-11
> **Author response**
>
> Thank you for your review. The main theoretical issue with using Radamacher complexity is (as you point out) data reuse. Because of data reuse, the relevant concept class for Radamacher complexity calculations would not be just the simple classifier on its own, but rather the composition of the simple classifier with the (complex) representation. The Radamacher complexity of this composed class will be very large, and so will not allow us to prove non-vacuous generalization bounds. Remark 3.2 in the paper gives an SSS algorithm where the generalization gap is arbitrarily large even though the downstream classifier is linear and has small Rademacher complexity.
>
> - We agree that the term “rationality” is not perfect, but we did not find an appropriate term that would capture this notion. If we think of the training algorithm as being used by an agent whose utility grows with the performance, then using an algorithm with a positive rationality gap is “irrational” in the sense that there is an alternative algorithm that can achieve better performance from the same training data.
> - You are right that SS+S is not the dominant methodology. We will make this clear. Throughout the paper we compared SS+S with end-to-end supervised training, since we did not want to introduce and discuss the intermediate SS+”fine tuning” model.  You are right that it is widely used in practice, and understanding the extent to which our results apply to it is an important open question.
> - You are right that this is a scatter plot. Because we sorted the points according to their generalization gap, and studied enough models for the graph to be dense, we thought it does make sense to draw lines and shades since it is possible to see some trends (eg the correlation between the generalization gap and the memorization gap), and it also makes it easier to visualize the different components of the RRM bound. However, we will edit the caption to make it clear that the shading is drawn for visual convenience only.
> - You are right that the RRM bound will hold even if we don’t take the maximum with zero. However, in this case, the rationality gap could sometimes be extremely negative, and so it will be less informative. Indeed, for standard interpolating classifiers the rationality gap is often very negative, while for SSS algorithms it is often only mildly so. Our assumption is that the rationality gap is only bounded from above by a small amount, which holds in both cases.
> - We will clarify remark 3.3 better. Generally, in the setting we are interested in, it may well be the case that after conditioning on x, there is no entropy left in y. Hence we cannot use standard information-theoretic bounds to argue about generalization. Instead, we use this “hypothetical experiment” in which we inject label noise, and argue about the ability of the simple classifier to fit this noise.
> - The paper includes an experiment and some discussion on the effect of changing eta in Appendix D.5. Generally, if we increase eta, then we expect the robustness gap to grow, and the memorization gap to decrease. In particular in our rigorous bound we lose a factor of 1/eta in the memorization gap bound. For the rationality gap, the effect is less clear (and indeed in many cases it is zero regardless).  For standard algorithms, we expect the robustness gap to never be more than eta, and so we want eta to be small enough so that we are OK with losing this factor, and large enough that the memorization gap is small.
> - We will expand on Section 2.1 to make it less dense.
> - Re Figure 2 - thank you!

---

### Official Review · AnonReviewer2 · 2020-10-30
**a new notion of generalization, with motivation and some experimental justification**

**Rating:** 7
**Confidence:** 4

**Review:**

This paper gives a new perspective on generalization, motivated by the success of self-supervised learning, especially on noisy data. They view the generalization error as consisting of 3 independent components: robustness, rationality and memorization. Informally, robustness measures the degradation in training accuracy due to the addition of noise. rationality measures the gap between noisy training and test accuracy and memorization is the gap between the training error on the uncorrupted and corrupted examples. The main point of the paper is that the memorization gap is smaller for self-supervised, simple algorithms compared to full supervised algorithms. They prove that when the noise is defined by a small fraction of labels being randomly flipped, then the memorization error of such simple algorithms can be bounded in terms of their information-theoretic complexity, independent of the representation they produce. (The proof is simple once the definitions are set up properly).
The most interesting part of the paper is an experimental study of several training procedures on benchmark data sets, measuring the three notions of error, matching what their theoretical result and overall motivation suggests. While perhaps not directly relevant to the practice of ML, this paper gives some explanation of the success of self-supervision in a toy setting, and is likely to inspire further research.
--- You study random label noise. What about attribute noise? And what is the noise is not random but adversarial? Do at least the notions still make sense?
--- the rationality gap is still confusing, and your dog/cat example does not help. More intuition/better examples would be great.

---

> ### Author Response · Authors · 2020-11-11
> **Author response**
>
> Thank you for your comments. We use the noise as a hypothetical experiment to obtain lower bounds on the noise-free setting, but the noise robustness of SSS algorithms is on its own an interesting subject to study, and we agree that different noise models would be important to look at.
>
> We agree that we should add more explanations and discussions to the rationality gap. Perhaps the best way to understand it is through the proof of the “performance on the table theorem”. This theorem holds regardless of the method of training - whether SSS or end-to-end supervised training. Consider the following example - suppose that there is an algorithm that, when trained with 5% label noise, manages to recover the original label for 90% of the datapoint for which it was given noisy labels, but only succeeds with 80% probability on new unseen test samples. In this case, we could obtain a procedure that achieves 90% test accuracy by doing the following - given the training set S and a fresh test sample x, add x into the training set with a random label y’. Since the model achieves 90% accuracy on noisy training points, with probability 90% we will recover the original label y for this fresh test sample. This would be computationally expensive, but it does demonstrate that having a positive gap between performance on noisy samples and test accuracy is “irrational” in the sense that you are achieving less performance than what could be achieved with the same data.

---

### Author Response · Authors · 2020-11-11
**Author response to high level points raised by reviewers**

We thank the reviewers for their careful reading of our work. We understand that there are multiple points where our exposition may be lacking, and we will take this into account in future revisions. We start by responding to high level points raised by the reviewers, and then comment separately on more specific points.

1. We respectfully disagree with AnonReviewer1’s claim that the paper is “over-claimed” and “non-rigorous”. Our main theoretical result is a rigorous proof of a theorem yielding a bound on the generalization gap of SSS algorithms under clearly-stated assumptions. These assumptions are highlighted in both the abstract and body of the paper, and hence we do not believe that we are “over claiming”. Specifically,  the assumptions used to obtain a bound on the generalization gap are that the rationality and robustness gaps are bounded. To our knowledge, all generalization bounds for deep learning are based on assumptions (see the related works section of [1] that reviews about 50 deep learning theory papers, discusses the assumptions used, and points out that they often do not hold in practice). Our assumptions do in fact (a) hold in practice (as demonstrated by our empirical study) and (b) do not “assume away the difficulty” since there are many examples, including all interpolating classifiers, where the assumptions hold (both rationality and robustness are small) but the generalization gap is large.

2. In addition to our theoretical contributions, we conduct the first extensive empirical study of SSS algorithms, demonstrating that they have small generalization gap, and showing that their robustness and rationality gaps are small. We also show that Zhang et al’s results on the ability of deep nets to fit random noise do not extend to SSS algorithms. We believe these empirical contributions are of independent interest.

3. Several reviewers wondered why our results do not follow immediately from prior bounds such as Radamacher complexity and others. This is a natural question to ask, given that the second-stage classifier is simple and has small capacity. The main issue is, as pointed out by AnonReviewer2,  the issue of data reuse. Generalization would indeed follow from standard bounds if we “held out” part of the data, did not use it for the representation learning, and only used it in the second stage to learn a classifier. However, without such a partition, generalization is not guaranteed. While (as we show) the generalization gap of SSS algorithms is small in practice, we can construct artificial examples of SSS algorithms where the classifier is simple (in particular linear with small capacity and small Radamacher complexity) but the generalization gap is high. (See Remark 3.2 in the paper, and we will include additional discussion on this in the updated version of the paper.)  This example also shows why some additional assumptions are in fact necessary to prove generalization bounds for SSS algorithms.

References:
[1] Arora, S., Du, S. S., Hu, W., Li, Z., & Wang, R.. Fine-grained analysis of optimization and generalization for overparameterized two-layer neural networks. ICML 2019 (pp. 477-502). See also arXiv preprint arXiv:1901.08584

---

### Decision · Program_Chairs · 2021-01-07
**Final Decision**

**Decision:**

Accept (Poster)

**Comment:**

The paper offers a new take on generalization, motivated by the empirical success of self-supervised learning.  Two reviewers found the contribution novel and interesting, and recommend acceptance (with one reviewer championing for it).  Two reviewers remain skeptical about the value of the paper, and the authors are encouraged to add a discussion about the points made in these reviews.

I agree with the positive reviewers and would like to recommend acceptance.